# CRITICAL SPARK: EVALUATING LLMS' ABILITY TO IDENTIFY INCONSISTENCIES IN PROBLEM FRAMING

## ABSTRACT

Large language models (LLMs) excel in complex tasks but often struggle with inconsistencies in problem framing, a critical skill for real-world scenarios. This paper introduces **SPARK**, a novel evaluation framework grounded in the Hierarchical Three-Space Theory, to assess LLMs' ability to identify missing information and challenge flawed problem setups. We propose a general framework to create benchmarks by introducing inconsistencies and misleading cues in diverse question-answering datasets, covering mathematics, science, and reading comprehension. To assist with robust measuring of critical thinking, we employ two key metrics: problem-solving capability rate and challenge rate. Our experiments with state-of-the-art LLMs reveal their limitations in critical thinking, particularly in recognizing inconsistencies. We also explore mitigation strategies, such as modified prompting and targeted fine-tuning. Furthermore, we conduct comprehensive experiments to investigate how model and problem properties influence critical thinking capabilities in LLMs.

## 1 INTRODUCTION

As large language models (LLMs) become increasingly integrated into decision-making processes, ensuring they possess robust critical thinking skills is of paramount importance. While significant attention has been given to LLMs' ability to generate responses and solve problems, the research community has also recognized the importance of understanding the limitations and potential risks associated with these models (Weidinger et al., 2022; Kaddour et al., 2023). A crucial question arises:

> *Can LLMs critically assess the very foundation of a problem—its initial framing—and identify inherent inconsistencies?*

Failure to do so could lead to flawed reasoning, inaccurate conclusions, and ultimately, unreliable performance, especially in complex, real-world scenarios.

Recent research has explored various facets of critical thinking in AI, including handling incomplete or ambiguous requests (Asai & Choi, 2021; Kamath et al., 2020; Kuhn et al., 2022), discerning truth from falsehood (Xu et al., 2023; Chen & Shu, 2023), and reconciling contradictory information (Xie et al., 2023; Zhou et al., 2023). However, the ability to recognize inconsistencies in problem framing remains under-explored. Current evaluation methods, while providing valuable insights into model performance on well-defined tasks, often fail to capture the challenges posed by such inconsistencies. This limitation highlights a significant gap in our understanding of LLMs' capabilities.

This study contributes to the ongoing discussion of LLM capabilities by introducing a novel framework for assessing this specific aspect of critical thinking in problem-solving. We contribute a general methodology for creating benchmarks to assess this crucial skill, addressing a significant gap in current evaluation methods. Our work is grounded in the Three-Space Theory of Problem Solving (Burns & Vollemeyer, 2000), which describes problem-solving as a process of interacting searches within three distinct but interconnected spaces: the Problem Framing Space (the general understanding of the task), the Strategy Space (possible solution approaches), and the Implementation Space (specific applications of those strategies).

DEFINITION OF CRITICAL THINKING FOR LLMS. Building upon the Three-Space Theory, critical thinking for LLMs is the ability to analyze the *Problem Framing Space* and recognize flaws in its definition, potentially by leveraging the *Strategy* and *Implementation Spaces*.

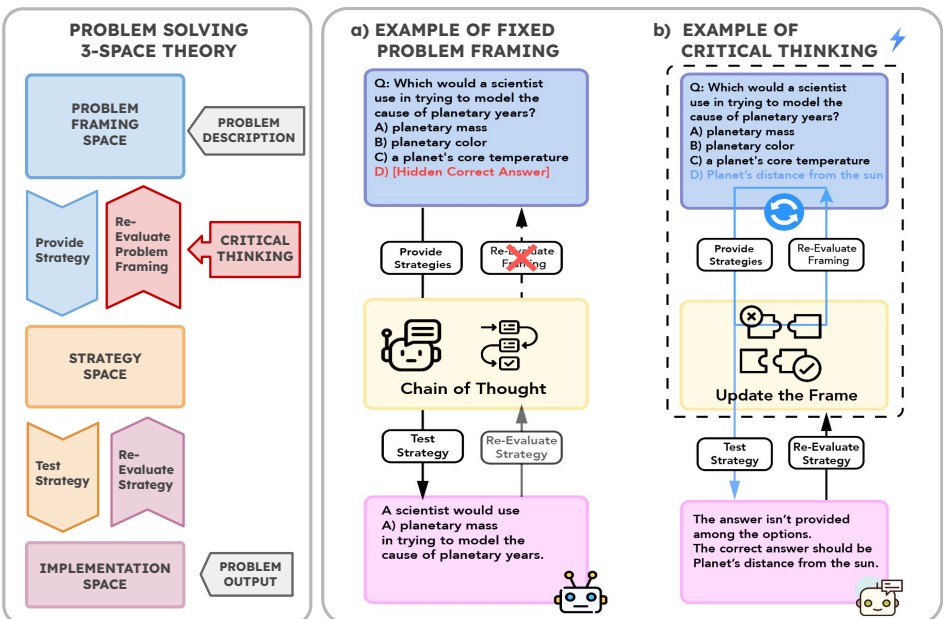

Figure 1: The Hierarchical Three-Space Theory of Problem-Solving adapted from Burns & Vollemeyer (2000), illustrating the interplay between Problem Framing, Strategy, and Implementation Spaces. Critical thinking involves recognizing flaws in one's understanding of a problem and leveraging feedback from the problem-solving process. The multiple-choice example illustrates this: an LLM, despite possessing relevant knowledge, might be constrained by a flawed problem setup, leading it to select an incorrect option and fabricate an explanation. However, an LLM with critical thinking capabilities would identify the issue and challenge the implausible options.

In this paper, we present a series of experiments designed to evaluate critical thinking in LLMs, focusing on their ability to recognize inconsistencies in problem framing and exploring these inconsistencies through the five key aspects outlined in our **SPARK** framework. These experiments encompass various dimensions, including assessing the impact of problem-solving strategies (SSI Hypothesis), examining the effects of problem complexity and misleading information (PSS and RMI Hypotheses), analyzing cross-domain generalization (ADA Hypothesis), and investigating the role of in-context learning and model training (KBC Hypothesis). While our study explores a breadth of problem types, we specifically focus on the LLMs' capacity to recognize when their initial problem model is insufficient and how they respond to new information or contradictions that challenge their initial understanding. This focus allows us to gain a deeper understanding of LLM reasoning and contribute to the development of models that can reliably handle complex, real-world scenarios.

The rest of this paper is organized as follows: Section 2 reviews related work in problem-solving, LLM evaluation, and critical thinking in AI. Section 3 details our theoretical framework, and describes our methodology, including benchmark creation and experimental design. Section 4 presents our experiments and results. Section 5 discusses the implications of our findings, and Section 6 concludes with a summary and directions for future work.

## 2 RELATED WORK

**Problem-Solving in Cognitive Science** The Hierarchical Three-Space Theory of problem-solving, which underpins our **SPARK** framework, is grounded in classic cognitive science theories (Newell, 1972; Stein et al., 1984) and addresses challenges of ill-structured problems (Rittel &

Webber, 1973; Simon, 1973). Its dynamic Problem Framing Space aligns with metacognitive processes (Flavell, 1979) and complex problem-solving research (Dörner, 1986; Funke, 2010; Greiff et al., 2014), representing interactions between problem framing, strategy development, and implementation. The theory integrates critical thinking skills (Elder & Paul, 2007; Facione, 1990; Dwyer et al., 2014) and resonates with current complex problem-solving (CPS) frameworks (Quesada* et al., 2005; Grable, 2006). SPARK extends these foundations, offering complementary perspectives for evaluating LLMs. It provides a structured approach to assess critical thinking in artificial agents, introducing quantifiable metrics like correctness, problem-solving capability and challenge rates. SPARK's examination of interactions between Problem Framing, Strategy, and Implementation spaces offers a novel lens for understanding complex problem-solving processes. The LLM problem-solving strategy is shaped by prompting techniques. Chain-of-Thought (CoT) (Wei et al., 2022) breaks down problems into intermediate reasoning steps. Tree-of-Thought (Yao et al., 2024) extends CoT by exploring multiple branches of reasoning through a tree structure. Graph-of-Though (Besta et al., 2024) extends CoT by structuring the reasoning process as a graph. Algorithm-of-Thought (Sel et al., 2023) provides well-defined rules to guide the LLMs to reason logically and effectively. By focusing on LLMs' robustness to misinformation and the influence of model architecture on problem-solving capabilities, SPARK addresses contemporary challenges in AI.

**Critical Thinking in AI** Recent literature explores critical thinking in AI through various lenses, including LLM noncompliance (Asai & Choi, 2021; Kamath et al., 2020; Brahman et al., 2024), misinformation susceptibility (Xu et al., 2023; Chen & Shu, 2023), knowledge conflicts (Xie et al., 2023; Zhou et al., 2023), input perturbations (Jia & Liang, 2017; Zhao et al., 2021), and sycophancy (Perez et al., 2023; Wei et al., 2023). These studies examine various facets of critical thinking in LLMs, including their ability to recognize limitations, handle misinformation, resolve contradictions, and resist biases. An emerging trend focuses on evaluating LLMs' ability to assess and correct reasoning processes, as exemplified by benchmarks like MR-BEN (Zeng et al., 2024), PRM800K (Lightman et al., 2023), and MR-MATH (Xia et al., 2024), and others that evaluate higher-order cognitive skills by examining the reasoning process. Recent work has further explored LLMs' capacity for self-correction (Tyen et al., 2023; Huang et al., 2023) and provided metrics for scoring step-by-step reasoning (Golovneva et al., 2023).

Our work distinguishes itself by focusing on LLMs' capacity to critique problem formulations across domains and actively identify flaws in problem setups, a fundamental aspect of critical thinking often overlooked. Using the Three-Space Theory, we provide a unified framework to evaluate this ability in multiple-choice, mathematical, and reading comprehension tasks, offering a comprehensive, cross-domain analysis of this critical thinking skill. Compared with existing benchmarks evaluating LLM on ambiguous or unanswerable questions (Brahman et al., 2024; Tian et al., 2023; Min et al., 2020), we create our dataset by modifying the options or context of well-defined questions and provide a more fine-grained analysis of LLM responses, investigating the factors that influence their critical thinking capabilities.

## 3 SPARK FRAMEWORK FOR CRITICAL THINKING IN LLMS

### 3.1 ADAPTING HIERARCHICAL THREE-SPACE THEORY FOR LLMS AND ESTABLISHING SPARK HYPOTHESES FOR CRITICAL THINKING

We adapt the Hierarchical Three-Space Theory (visualized in Fig. 1) to the context of language model processing, reframing the three spaces as:[1]

**Problem Framing Space** (Model Space): Represents the LLM's initial understanding and assumptions about the given task/question, derived from the prompt and the model's pre-trained knowledge.

**Strategy Space** (Hypothesis Space): Encompasses potential reasoning paths or approaches to address the task, manifesting in the model's generation of intermediate thoughts or steps, such as those observed in chain-of-thought reasoning. The exploration of this space is influenced by the model's training and the specific prompting technique used.

**Implementation Space** (Experiment Space): Represents the actual output generation process, including token-by-token text generation where the model applies its selected strategy to produce a response. This space is directly observable through the model's output.

---

[1]We have renamed the spaces from the original Three-Space Theory (shown in parentheses) to better reflect their application to LLMs and avoid terminological confusion (e.g., 'hypothesis space' or 'model' in ML).

Grounded in the Three-Space Theory, we analyze LLM critical thinking as an ability to facilitate feedback from the (Strategy and Implementation Spaces) to enable revisions in the Problem Framing Space. Building upon this adapted theory and the critical thinking definition, we propose the **SPARK** framework to evaluate the five key hypotheses to evaluate the critical thinking in LLMs:

**S**trategy Space Interaction (SSI) Hypothesis: The way LLMs solve problems (their Strategy Space) influences their ability to update their Problem Framing Space. To evaluate this hypothesis, we compare different solving strategies (Strategy Space), in particular original prompting vs chain-of-thought prompting, and evaluate the effect on the Problem-Solving Space (Section 4.3).

**P**roblem Space Sensitivity (PSS) Hypothesis: LLMs can detect inconsistencies or missing information in their Problem Framing Space, but this ability varies based on problem complexity and model architecture. To study the model's ability to detect insconsistencies or missing information of the problem statement, we first design problem setup to incorporate such cases. Then we evaluate the model's general ability to challenge the incorrect problem formulation (Section 4.1. We then study whether the degree of challenging depends on the model's solving capability for the given problem (Section 4.2) or the problem's complexity, where we simulate by increasing the number of missing constraints (Section 4.4).

**A**cross-Domain Abstraction (ADA) Hypothesis: LLMs' critical thinking abilities are partly domain-general, but effectiveness varies across problem types. To find the domain-general critical thinking ability of an LLM, we search over a wide range of diverse domain datasets to find the clusters of datasets that share similar critical-thinking patterns between datasets (Section 4.6).

**R**obustness to Misleading Information (RMI) Hypothesis: LLMs' Problem Framing Space can be influenced by misleading or noisy information. To test the robustness of the model to misleading information, we design experiments with conflicting information as a form of gaslighting hints in the prompt (Section 4.5).

**K**nowledge and Behavior Conditioning (KBC) Hypothesis: LLM knowledge and behavior that governs the dynamic interplay among all three spaces can be shaped or conditioned through fine-tuning and in-context learning. To investigate the interplay among all spaces, we study how different fine tuning strategies and datasets condition the model behavior (Section 4.7 and also how different in-context learning examples can affect the model critical-thinking ability differently (Section 4.8).

This framework with proposed experiments allow us to systematically investigate critical thinking in LLMs, providing insights into their capabilities and limitations in complex problem-solving scenarios nad how large language models (LLMs) navigate within the spaces of the Three-Space Theory and interact across them.

## 3.2 BENCHMARK CREATION OVERVIEW, REPRODUCTION, AND EXPERIMENTAL SETUP

Our work provides a framework to evaluate critical thinking of a large language model by modifying existing, correctly annotated datasets of interest.

**Datasets.** In our study, we employ multiple existing datasets covering a range of topics and skills: **8 multiple-choice datasets (QA)**: Hellaswag (commonsense NLI), TAL (math), OpenBook QA (text comprehension with commonsense reasoning), ARC Challenge (science), GPQA (domain-specific science), LSAT (law reading comprehension), MMLU-Math(math subset of general knowledge), TruthfulQA (human falsehood), **3 free-form generation datasets (FG)**: GSM8K (math), Quail (reading comprehension), and HotPotQA (multi-hop reasoning). For each dataset, we sample 300 test queries for evaluation. We refer the reader to Appendix A for further details on the construction of these datasets.

**Dataset Modification.** We create two new versions of these datasets to test LLMs' ability to detect inconsistencies or missing information in problem setups:

- **(Hidden Correct Answer)** For 8QA datasets, we remove the correct answer choice from the multiple answer choices. Here, we study whether the model is able to update its Problem Framing assumptions that the correct answer choice might actually not be provided within the problem statement. Thus, requiring the model to change its own initial assumptions about the multiple-choice problems.

- **(Missing Information)** For 3FG datasets, we remove the necessary condition from the problem statement so that the answer cannot be inferred from the provided context, thus, requiring the LLM to update its Problem Framing Space assumptions that the model cannot arrive at the final answer due to missing information. The detailed question modifications are explained in A.1.2.

These modifications allow us to evaluate the model's ability to recognize inconsistencies and challenge insufficient problem setups. Crucially, we assess the model's capacity to self-recognize these flaws without any additional guidance. These datasets span diverse problem types—including mathematics, reading comprehension, domain-specific science, and story completion—each designed to evaluate specific problem-solving skills. We prioritize reasoning tasks as they align with our definition of critical thinking, while providing observable intermediate steps that enable us to evaluate inconsistencies in LLMs' inference processes. To further evaluate robustness to misleading information, we augment the 8QA datasets by creating three versions with different misleading hints:

- **(Gaslight Correct)**: By the end of each problem statement, we add a hint claiming that the correct answer (e.g., *9*) is incorrect (e.g., *Hint: 9 is incorrect*).
- **(Gaslight Wrong)**: By the end of each problem statement, we add a hint claiming that the wrong answer (e.g., *8*) is correct (e.g., *Hint: 8 is correct*).
- **(Gaslight Both)**: By the end of each problem statement, we add a hint claiming that the wrong answer is correct and the correct answer is incorrect (e.g., *Hint: 8 is correct and 9 is incorrect*).

**Models.** After building the evaluation datasets, we aim to evaluate LLMs across a range of training parameter sizes and diverse capabilities. Therefore, we include the following models: Llama-3.1-8/70B-Instruct (Dubey et al., 2024), Mistral-7B-Instruct-v0.3, and GPT4o (Achiam et al., 2023). We configure each model with a temperature of 0 and a maximum token limit of 1024 for inference. For more model, inference, and training details, we refer the reader to Appendix B.

**Evaluation Metrics.** In all experiments, we measure two key metrics for critical thinking evaluation:

- **Problem-Solving Rate:** Measures whether the LLM's incorporates the correct knowledge about the question We leverage binary correctness label on clear generative tasks $cor_c$ and modified questions $cor_m$, where the correctness represents whether the response demonstrates the correct knowledge. To construct clear tasks, we remove options for the multiple-choice problem and use the original questions for free-form generation problems. Problem-solving capability is measured by $cor_c \cup cor_m$, as correct solutions in either scenario indicate the model's ability to solve the task.
- **Critical-Thinking Rate:** Measures the LLM's ability to identify flaws in the problem setup. We first identify well-defined questions that the LLM does not challenge the problem setup. Let $N_1$ denote the number of unchallenged clear questions, and $N_2$ denote the number of their corresponding modified versions that are challenged. The ratio $\frac{N_2}{N_1}$ measures the LLM's capability to identify problem inconsistencies. The detailed explanation is in Appendix C.

We employ off-the-shelf LLMs to measure these two scores for efficient evaluation. Particularly, we use Llama-3.1-70B-Instruct to measure the correctness of the answer with respect to the ground truth answer and GPT-4o to measure the challenge rate of the models. Due to high efficacy, we choose these models as the judges, reaching 100% accuracy in measuring correctness and 92% accuracy in measuring challenges on our manually curated held-out datasets, respectively. We provide relevant template judge prompts in Appendix D.

**Full Reproduction.** To assist readers with reproduction of our study, we publish the codes for tuning and inference, (hold-out) datasets, and full responses (`https://anonymous.4open.science/r/Critical-Spark-6EE3/`).

# 4 EXPERIMENTS AND RESULTS

Now, we present our analysis on each experiment delineated in Section 3 and study the relation to critical thinking ability. Due to space limitations, we move most of our figures and numerical tables to Appendix E, while keeping the summarized results and analysis in the main text.

## 4.1 ABILITY TO CHALLENGE ASSUMPTIONS

We analyze LLMs' critical-thinking rate defined in Sec 3.2 using problems lacking the correct option or key information. Figure 2 shows that all models demonstrate this capability across the studied datasets. For multiple-choice problems, the highest challenge rates (22-27%) are observed on MMLU, TAL and TruthfulQA, which are primarily mathematical and factual datasets. For free-form generation tasks, larger models such as GPT-4o and Llama-70B achieve around a 75% chal-

lenge rate, indicating their proficiency in identifying inconsistencies in these math problems. Furthermore, Mistral-7B-Instruct-v0.3 and GPT-4o challenge assumptions most often across datasets; however, since all prompts contain missing information, the current levels of challenge rates are still far below the expected 100%, indicating that while LLMs possess some critical thinking ability, there is significant room for improvement. While LLMs demonstrate a capacity to challenge assumptions, their proficiency appears to be influenced by dataset characteristics, model scale, and instruction-following training, as suggested by the PSS hypothesis.

## 4.2 SOLVING VS CHALLENGING CAPABILITY

We investigate the relationship between problem-solving ability (correctness rate on complete problems) and critical thinking (challenge rate on incomplete problems). Figure 2, 14 reveals *no clear correlation* between these two abilities, suggesting these may be distinct skills potentially influenced by factors such as dataset characteristics, model architecture, and prompting. This aligns with the PSS hypothesis, as it demonstrates that the ability to challenge inconsistencies is not solely dependent on problem-solving proficiency. GPT-4o and Llama-70B exhibit high performance in both problem-solving rates and critical-thinking rates on GSM8k. While Llama-70B achieves better problem-solving performance on OpenbookQA, it shows lower critical thinking rates compared to GPT-4o. Mistral-7B, despite having the lowest problem-solving rate on TAL, maintains a relatively high critical thinking rate. The Problem Framing Space can be updated even when the model cannot solve it.

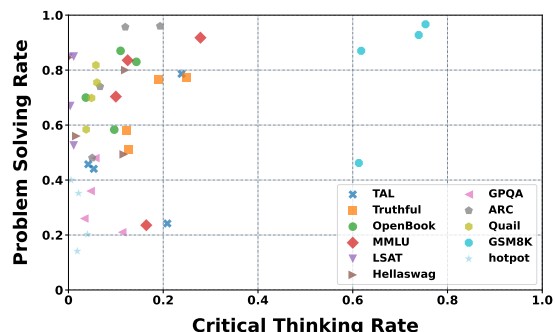

Figure 2: **Problem-Sovling vs Critical-Thinking Across Datasets and Models.** Each data point represents a specific dataset (indicated by shape). The performance is evaluated across GPT-4o, Llama-3.1-7bB, Llama-3.1-7B and Mistral-v0.3-7B. This visualization highlights the variation in correctness and challenge rates across different tasks

## 4.3 IMPACT OF PROBLEM-SOLVING STRATEGIES

We investigate the impact of CoT strategy on critical thinking capability. Figure 3 reveals *mixed results*. While CoT increases critical thinking rates for Mistral-7B-Instruct-v0.3 in most cases, other models show notable decreases on TruthfulQA and Quail. On HypotQA, CoT improves problem-solving performance across all models, while slightly hindering problem-solving capabilities on MMLU. This variation may be attributed to increased cognitive load from generating and processing intermediate reasoning steps, or potential bias toward solution generation induced by CoT prompting ( see Sweller (1988); Evans (2003) for some

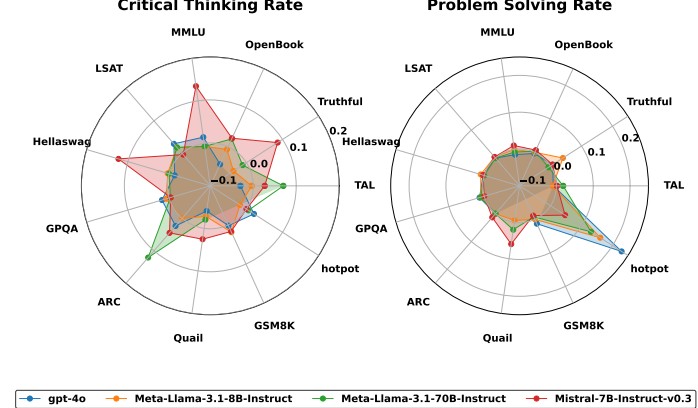

Figure 3: **Impact of CoT Prompting on Challenge and Correctness Rates.** The radar plot shows the difference in challenge rates(left) and correctness rates(right) between CoT prompting and original prompting across various datasets and LLMs. Positive values indicate improvement with CoT.

cognitive evidence). Additionally, the subtle variation in critical thinking performance on HotpotQA indicates that better problem-solving capability does not lead to more critical thinking. Dataset characteristics likely influence CoT's effectiveness, as problem representation affects problem-solving strategies (c.f., Chi et al. (1981)). These observations highlight the nuanced nature of the SSI hypothesis, demonstrating that while Strategy Space modifications can influence the Problem Framing Space, the effects are multifaceted and not always predictable.

## 4.4 EFFECT OF PROBLEM COMPLEXITY

We investigate the effect of problem complexity, specifically the number of missing constraints in the GSM8K dataset, on LLMs' ability to challenge assumptions. Figure 4 shows that increasing the number of missing constraints generally increases the challenge rate, with Mistral-7B-Instruct-v0.3 reaching 89% when three constraints are missing. When presented with a clearly stated question, LLMs tend to frame it as a mathematical problem, approaching it step-by-step to arrive at a numerical result. However, as we progressively remove necessary conditions from the question, LLMs increasingly adopt a more critical approach, focusing on evaluating the problem's solvability rather than directly generating a solution. This shift prompts them to consider the question's solvability, leading to an increased rate of challenge to the problem's premises. This aligns

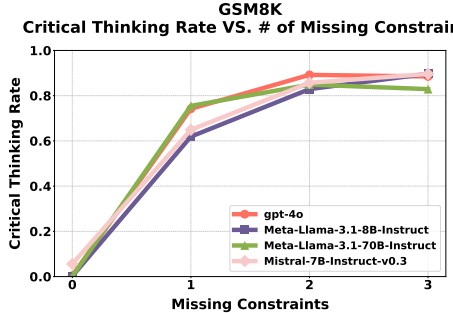

Figure 4: The impact of varying the number of missing constraints on the assumption rate.

with the PSS Hypothesis, which suggests that LLMs' sensitivity to inconsistencies is influenced by problem complexity. However, it's important to acknowledge that our automatic evaluation template (Appendix D), while achieving 95% accuracy, might not perfectly capture the nuances of LLMs' challenge responses, potentially contributing to the observed variations.

## 4.5 ROBUSTNESS TO MISLEADING INFORMATION

| | Correctness Rate Change | | | | Challenge Rate Change | | | |
|---|---|---|---|---|---|---|---|---|
| | | | ARC Dataset | | | | | |
| Model | Original -
Gaslight Correct | Original -
Wrong | Original -
Both | Average | Gaslight Correct
-Original | Wrong
-Original | Both
-Original | Average |
| Llama-3.1-70B-Instruct | 0.42 | 0.76 | 0.85 | **0.68** | 0.03 | 0.17 | 0.05 | **0.08** |
| Llama-3.1-8B-Instruct | 0.16 | 0.14 | 0.24 | **0.18** | 0.06 | 0.07 | 0.09 | **0.07** |
| Mistral-7B-v0.3-Instruct | 0.41 | 0.51 | 0.63 | **0.52** | 0.03 | 0.09 | 0.04 | **0.05** |
| gpt-4o | 0.30 | 0.70 | 0.68 | **0.56** | 0.09 | 0.22 | 0.17 | **0.16** |
| | | | TAL Dataset | | | | | |
| Llama-3.1-70B-Instruct | 0.21 | 0.21 | 0.31 | **0.24** | 0.04 | 0.08 | 0.10 | **0.07** |
| Llama-3.1-8B-Instruct | 0.10 | 0.11 | 0.21 | **0.14** | 0.07 | 0.07 | 0.09 | **0.08** |
| Mistral-7B-v0.3-Instruct | 0.16 | 0.16 | 0.21 | **0.18** | -0.09 | 0.05 | -0.05 | **-0.03** |
| gpt-4o | 0.17 | 0.35 | 0.42 | **0.31** | 0.11 | 0.22 | 0.32 | **0.22** |

Table 1: Impact of Misleading Information on Correctness and Challenge Rates in ARC Challenge.

We study the robustness of LLMs' critical thinking by introducing misleading information ("gaslighting") into the ARC Challenge dataset. We append misleading hints after the problem description to introduce inconsistency into the original problem setup. We measure both challenge and correctness rates across three gaslighting conditions (see Table 1). While gaslighting increases the challenge rate across all models, it simultaneously decreases the correctness rate (Table 1). These findings are consistent across other datasets (see Appendix E). We observe that misleading hints can influence LLMs to select incorrect options, decreasing the correctness rate. When generating inference steps to support their wrong choices, the LLMs produce reasoning paths that contain counterfactual or flawed statements. The increased challenge rate in these cases suggests that when reasoning paths contain obvious errors or contradict common sense, LLMs are more likely to identify inconsistencies and challenge the problem setup. This demonstrates that LLMs exhibit critical

thinking capabilities when the implausibility of their inference steps is obvious. LLMs can be robust against misleading hints, as their critical thinking capabilities enable them to challenge provided information.

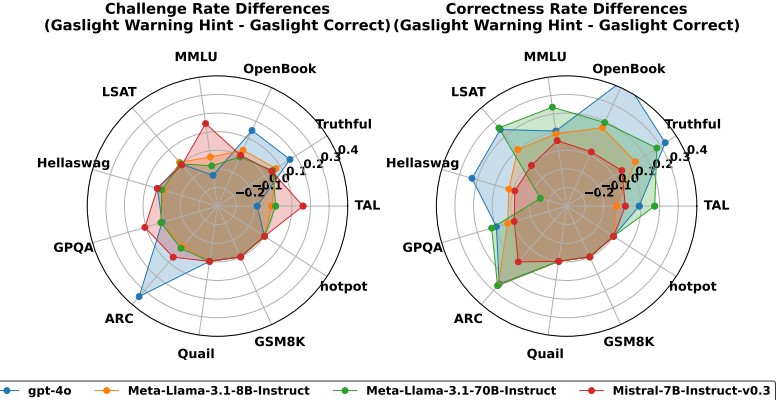

Figure 5: **Effect of Warning Hints on LLMs' Response to Misleading Information.** The radar plot shows the difference in challenge rates (left) and correctness rates (right) when LLMs are provided with a warning about potential misleading information, compared to no warning. Positive values indicate improvement with the warning hint.

We investigate whether warning LLMs about potential misleading information can mitigate its negative effects. Figure 5 shows that adding a warning hint maintains or increases challenge rates in many cases, while notably improving correctness rates across several datasets (with the largest improvement on OpenbookQA). This suggests that warning hints enable LLMs to better discern and resist misleading information, thereby improving their critical thinking.

### 4.6    CROSS-DOMAIN ANALYSIS

We study whether the ability to update the Problem Framing Space is similar across datasets or is domain-specific. To do so, we compare all 8 QA datasets with hidden correct information and compute the correlation between them across 4 models using the challenge rates adjusted by the correctness rate. From the correlogram in Figure 6, we can find highly correlated datasets, TAL with MMLU-Math (0.81) being the most significant, and OpenbookQA with GPQA (0.58) being the second most significant. Since, these datasets are mathematically and scientifically focused problems, this demonstrates the cross-domain ability of the models when the domains share some similarity. This shows that the ability to update the Problem Framing Space is consistent across different datasets with similar domains and aligns with the ADA Hypothesis.

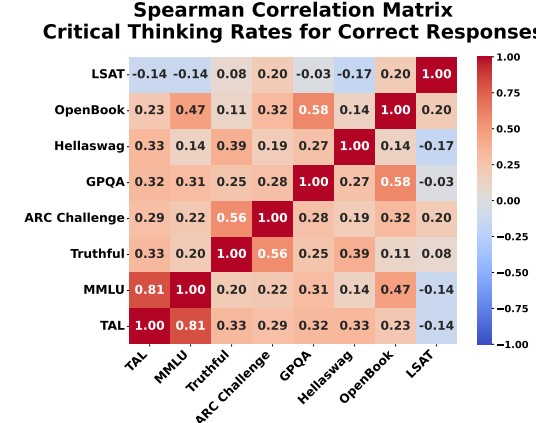

Figure 6: **Correlation of Critical-thinking Rates Across Datasets.** This correlogram displays the correlation coefficients between challenge rates on different datasets, after adjusting for correctness rates. Higher correlation indicates greater consistency in LLMs' critical-thinking capability across those datasets.

### 4.7    IMPACT OF FINE TUNING ON CRITICAL THINKING

We examine how fine-tuning affects the model's ability to challenge the problem. In particular, we look at supervised fine-tuned and human preference-tuned models and measure their challenge rates on the TAL dataset. We observe in Figure 7 that the safety instruction-following tuned Llama-3.1-8B-Instruct model on HH achieves a

lower correctness rate than the base model probably due to the HH dataset not being focused on the maths. Additionally, since the models are trained to follow instructions, they are also less capable of challenging when misleading information is provided, getting a lower challenge rate than what the base model achieved.

The LLM directly fine-tuned on the TAL achieves the lowest correctness, likely due to overfitting which impairs its ability to critically handle misleading information. The Llama-3.1-8B-Cobalt model achieves the best performance on both correctness and challenge rates. This success can be attributed to its training on a dataset five times larger than GSM8k and including more detailed and comprehensive reasoning steps. This training data encourages the model to generate logical inference steps and leverage intermediate reasoning to update its problem understanding. This suggests that tuning models with data, rich in reasoning steps, can improve the critical thinking ability of the model, demonstrating the interplay between all three spaces that agrees with the KBC Hypothesis. We provide details on the fine-tuned models in

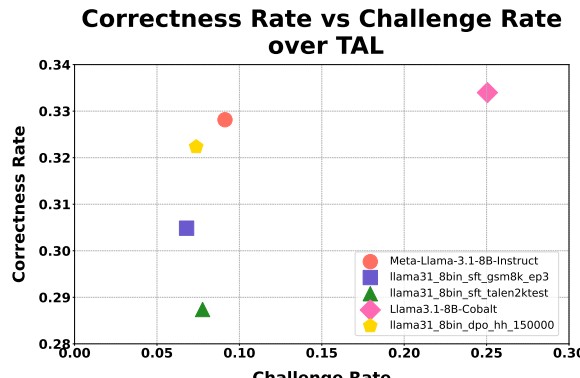

Figure 7: Correctness vs Challenge Rates for TAL across Llama-3.1-8B-Instruct fine-tuned models when faced with misleading information (gaslight correct).

Appendix B.1.1. We report consistent results on remaining gaslighting cases in Appendix E.

## 4.8 IN-CONTEXT LEARNING AND CRITICAL THINKING

We explore how in-context learning prompting affects the ability to update the Problem Framing Space. In particular, we measure the correctness and challenge rates when the model is provided with 3 examples in the prompt for 8QA datasets (QA_3_incorrect or 3-ICL). In Figure 8, we can observe a trend across models. In particular, the correctness rate when provided with in-context learning examples is similar to or even better than the correctness rate when no examples are provided. This suggests that having similar examples can better update the Problem Framing Space to suggest better strategies focused on similar types of problems

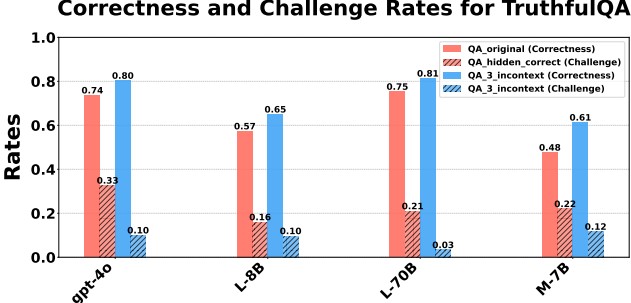

Figure 8: Correctness vs Challenge Rates for in-context learning on the TruthfulQA dataset across models. gpt-4o for GPT-4o, L-8B for Llama-3.1-8b-Instruct, L-70B for Llama-3.1-70b-Instruct, M-7B for Mistral-7b-v0.3

to correctly solve the problems. On the other hand, in-context learning struggles with missing information as the challenge rate has decreased across all models, which suggests that in-context learning can limit the critical thinking ability of the model, which agrees with the KBC Hypothesis. We observe similar trends on the remaining datasets and report all results in Appendix E.

While we observed that having three in-context learning examples can decrease the challenge rate, adding more in-context learning examples (from 3 to 5) will not fix that either. As we observe in Table 2, the challenge rates for three and five in-context learning examples (5-ICL) are close to each other as well as the correctness rate. One possible way to help the model to challenge assumptions is to provide examples of such action. Thus, when having examples of challenging assumptions in the context (5-ICL-C), we observe that for most of the models (gpt-4o, Llama-3.1-8B-Instruct, and Mistral-7B-Instruct-v0.3), the challenge rate is increased while the correctness rate is preserved. This experiment suggests ways to help the model improve its critical thinking through in-context

Table 2: Correctness vs Challenge Rates for in-context learning on the TAL dataset across models with varying number of examples and varying types of examples, including examples demonstrating challenging the assumptions. Performance across different ICL formats

| Model | Correctness Rate | | | Challenge Rate | | |
|---|---|---|---|---|---|---|
| | 3-ICL | 5-ICL | 5-ICL-C | 3-ICL | 5-ICL | 5-ICL-C |
| gpt-4o | 0.77 | 0.74 (↓ 0.03) | 0.73 (↓ 0.04) | 0.03 | 0.02 (↓ 0.01) | 0.06 (↑ 0.03) |
| Meta-Llama-3.1-8B-Instruct | 0.43 | 0.46 (↑ 0.03) | 0.40 (↓ 0.04) | 0.03 | 0.04 (↑ 0.01) | 0.02 (↓ 0.01) |
| Meta-Llama-3.1-70B-Instruct | 0.03 | 0.62 (↑ 0.59) | 0.62 (↑ 0.59) | 0.37 | 0.03 (↓ 0.34) | 0.03 (↓ 0.34) |
| Mistral-7B-Instruct-v0.3 | 0.25 | 0.26 (↑ 0.01) | 0.20 (↓ 0.05) | 0.12 | 0.11 (↓ 0.01) | 0.24 (↑ 0.11) |

learning examples, which shows that we can condition the LLM knowledge and behavior using appropriate examples, accepting the KBC Hypothesis.

# 5 DISCUSSION AND IMPLICATIONS

**Key Findings.** Our experiments reveal that while state-of-the-art LLMs demonstrate some capacity for critical thinking, their ability to consistently recognize and challenge inconsistencies in problem framing remains limited, as evidenced by the generally low challenge rates. These findings offer a nuanced understanding of the SPARK hypotheses. The PSS hypothesis is supported by the observation that larger models and those with instruction-following training exhibit higher challenge rates, but the overall low rates highlight the need for further research. The SSI hypothesis is supported by the mixed effects of chain-of-thought prompting, suggesting a complex interplay between strategy and problem understanding. The RMI hypothesis is confirmed by the observation that gaslighting increases challenge rates but reduces correctness, underscoring LLMs' vulnerability to manipulation.

**Implications for LLM Evaluation and Development.** These findings have implications for LLM development and evaluation. Our research underscores the need to incorporate critical thinking as a key evaluation criterion, using frameworks like SPARK to systematically assess these capabilities. For LLM developers, our findings highlight the need to explicitly incorporate critical thinking skills into model training and design, including enhancing robustness to misleading information, promoting deeper understanding, improving inconsistency detection, and optimizing prompting strategies.

**Limitations.** Our results span a diverse range of benchmark datasets, yet this selection is not exhaustive. Researchers can apply our evaluation methodology to their own datasets of interest to assess an LLM's critical thinking abilities. While our current evaluations focus on the final response output generated by LLMs, future work could delve deeper by analyzing model activations. Recent advancements in LLM reasoning have led to improved capabilities, as demonstrated by the gpt-o1 model. Due to its recent release, we have not had the opportunity to evaluate this model in depth. However, preliminary results suggest that even this advanced model may also face challenges in critical thinking tasks.

# 6 CONCLUSION AND FUTURE WORK

This paper presents a novel framework for evaluating critical thinking in LLMs, grounded in the Three-Space Theory. Our findings reveal limitations in LLMs' ability to challenge problem setups and highlight the influence of various factors (e.g., solving capability, problem complexity, misleading information, fine-tuning, and in-context learning) on their critical thinking capabilities. The proposed framework is readily adaptable across diverse problem types, providing a key step towards evaluating and enhancing critical thinking in LLMs.

Future research could extend this framework to more complex, real-world-oriented tasks like dialogue generation and code design. Additionally, our observations reveal that various prompting techniques including gaslight, gaslight with warning, and CoT influence the model performance, and we noted a trade-off between response correctness and critical thinking capability. Future work could investigate how to optimize this trade-off.

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

# A DETAILS ON DATASETS

## A.1 DATASETS FOR EVALUATING LLMS

- **Free Generation**

  1. GSM8K (Cobbe et al., 2021) includes multi-step, grade-school-level arithmetic problems designed to test LLMs' mathematical reasoning abilities. Each problem contains multiple necessary conditions, enabling us to quantitatively modify questions by selectively hiding a certain number of these conditions.

  2. HotpotQA (Yang et al., 2018) is a multi-hop reasoning dataset that challenges models to combine information from multiple documents. It provides several context documents, with only a few containing relevant information. LLMs must first identify these related documents before inferring the answer. The dataset includes titles of the related documents, facilitating quantitative modification by selectively omitting certain documents.

  Both GSM8K and HotpotQA require models to infer answers by leveraging multiple conditions in the context and following several reasoning steps. The multi-hop reasoning paths exhibited in these datasets effectively display LLMs' problem-solving strategies. Consequently, we can investigate how the LLM updates its Problem Framing Space assumption according to the feedback from the Implementation Space.

- **Multiple-Choice**

  1. QuAIL (Rogers et al., 2020) is a reading comprehension dataset that includes a variety of question types to evaluate a model's ability to evaluate models' abilities to understand and reason about complex texts

  2. TAL (matheval.ai, 2023) contains mathematical competition questions across primary, junior high, and high school levels. Each question requires multiple intermediate steps to solve, thus reflecting logical reasoning and critical thinking capability.

  3. TruthfulQA (Lin et al., 2021) is a dataset designed to evaluate an LLM ability to avoid generating false or misleading responses. It adversarially crafted problems that exploit common human misconceptions, potentially leading to incorrect answers. The dataset encompasses a diverse range of topics, including health, law, finance, and politics. By challenging models with questions that humans might answer incorrectly, TruthfulQA evaluates the LLM capacity to avoid generating false and misleading information.

  4. OpenBookQA (Mihaylov et al., 2018) includes elementary-level science questions, which require the LLM to retrieve of factual information and leverage external "open-book" knowledge to infer the answer. This dataset evaluates the LLM capability to integrate the retrieved information and logical inference.

  5. MMLU (Hendrycks et al., 2020) includes a wide variety of science questions from high school to professional difficulty levels. The dataset is designed to evaluate the understanding and reasoning capability of LLM across different domains of knowledge, thus displaying the depth and width of LLM understanding.

  6. GPQA (Rein et al., 2023) includes highly challenging, domain-specific questions across scientific fields such as biology, physics, and chemistry. The extremely difficult questions are used to evaluate the LLM performance on complex scientific inquiries. By using exceptionally difficult questions that are not tailored to any single discipline, GPQA evaluates the LLM's versatility and adaptability.

  7. AI2 Reasoning Challenge(ARC) (Clark et al., 2018) contains grade-school level science exam problems with two components: Easy Set and Challenge Set, where the questions from Challenge Set cannot be answered using simple fact retrieval or superficial reasoning. ARC evaluates the LLM's ability to understand scientific knowledge and integrate multiple information.

  8. LSAT (Zhong et al., 2021) includes standardized questions primarily used for law school admissions focusing on logical reasoning and reading comprehension skills. These complex tasks, set within law-related contexts, are designed to evaluate an LLM's analytical reasoning and cognitive abilities.

  9. HellaSwag (Zellers et al., 2019) contains multiple-choice questions that challenge language models to select the most plausible continuation of a given scenario. This prediction task evaluates the LLM capability of commonsense reasoning, context understanding, and

logical inference. HellaSwag assesses their ability to grasp implicit information, apply real-world knowledge, and make sensible deductions.

For the multiple-choice dataset, the LLM needs to integrate diverse information and navigate multiple inference steps to arrive at a result. The crafted options in these datasets provide an effective means to evaluate an LLM's critical thinking abilities. This format operates under the initial assumption in the Problem Framing Space that the correct answer is among the provided options. While removing the ground-truth option, we may observe the LLM generate the correct answer during its reasoning process, yet still select an incorrect option from those provided. Alternatively, it might recognize that all given choices are unsuitable, but nevertheless feel compelled to choose one. These scenarios clearly demonstrate how an LLM can be constrained by its initial assumptions, revealing a reluctance or inability to update its framework when faced with conflicting evidence.

### A.1.1 DATASET CONSTRUCTION

We select 300 examples from the training set and 3 from the test set for in-context prompting. For datasets with fewer than 300 examples, we subtract 3 from the training set to ensure we have the necessary test examples. This process allows us to maintain a balanced in-context learning setup, where the model uses the selected examples to infer patterns and generalize to new data points. Even for smaller datasets, we ensure consistent evaluation by prioritizing a balance between training and test examples without compromising the in-context learning tasks, such as QA_3_incontext.

### A.1.2 MODIFICATION OF GENERATIVE TASKS

We constructed generative tasks using four established datasets: GSM8k, HotpotQA, and QuAIL. To evaluate critical thinking capabilities, we deliberately introduced inconsistencies that make these problems unsolvable.

- GSM8k contains arithmetic problems, where the final answer is calculated by all the numerical conditions provided in the context. We design a reliable template to leverage GPT-4o to rephrase the problem context and remove one provided numerical condition.
- HotpotQA is a multi-hop reasoning task, requiring information extraction from multiple documents. The dataset provides the indices of related documents and sentences. We create incomplete tasks by removing one relevant document from the required set
- Quail is a reading comprehension dataset and includes questions whose correct answer is "not enough information". We directly sample some questions and corresponding paragraphs as incomplete reading comprehension tasks

Template for removing numerical conditions from GSM8k questions. The modified questions are generated by GPT-4o through this template.

> Consider this math problem, can you rephrase the problem context and hide one condition, which is provided numerically? Remember only hide one condition and keep the left numerical values.
> **Question**: John bought a T-shirt for $10, a pair of shoes for 20$. How much does he spend?
> **Rephrased Question**: John bought a T-shirt for a certain amount of money, a pair of shoes for 20$. How much does he spend?
> **Question**: (The question requires modification)
> **Rephrased Question**:

### A.1.3 MODIFIED MATH PROBLEM

We manually select the validation dataset to test the accuracy of the automatic template. We collect 100 ground-truth responses for each category: 1. response directly saying the answer cannot be determined. 2. reference answer solving the answer fluently and providing the numerical result. 3. response assigning a variable for the missing condition and providing the formula. The accuracy for each type of validation data is 0.971, 1.00, and 0.957.

**Input:** Modified math question and LLM response.
We define three categories for the response. Type 1 and Type 3 are considered to show critical thinking capability.

**Prompt:**
Consider this modified math problem with one condition removed: Modified Question. Here is the LLM response: LLM response.
Consider these different categories, which one do you think describes the sampled answer best?
1. The response directly mentions the answer can not be determined due to the missing information. The response does not provide the calculation.
2. The response solves the question and provides the numerical result as if all necessary conditions are provided.
3. The response provides a formula including the unknown variable during inference.
Directly answer with your choice number

Template for removing constraints. (Missing Constraints)

**Original Question:**
Harper drinks a 1/2 bottle of mineral water per day. She buys the mineral water by the case at a big box store. They come 24 bottles to a case and are currently on sale for 12.00. If she wants to buy enough cases to last her 240 days, how much will she spend?

**Prompt:**
Consider this math problem, can you rephrase the problem context and hide one condition, which is provided numerically? Remember only hiding one condition and remain the left numerical values.
Question: Tom has 5 apples. Ann has 4 apples. How many apples in total?
Rephrased Question: Tom has 5 apples. Ann has a certain number of apples. How many apples in total?
...(simple examples)
Question: (original question)...
Rephrased Question:

**Modified Question:**
Harper has a daily habit of consuming half a bottle of mineral water. She prefers to purchase her supply in bulk from a large retailer. The store offers a case of these bottles, each containing a *certain number* of bottles, for a price of 12.00. Harper plans to stock up enough cases to last her for 240 days. How much money will she need to spend?

## B    DETAILS ON MODELS

### B.1    MODEL INFORMATION

We have evaluated on the following LLMs to cover a wide range of training parameter sizes as well as diverse capabilities: Llama-3.1-8/70B-Instruct (Dubey et al., 2024), Mistral-7B-Instruct-v0.3, and GPT4o (Achiam et al., 2023).

### B.1.1    FINE-TUNED MODELS

In Section 4.7, we evaluate the performance of the Llama-3.1-8B-Instruct on the challenging mathematical dataset, TAL, under the gaslighting setting. Observing, low correctness rate of the original

model on the test TAL dataset, we study how fine-tuning affects the ability of the model. We evaluate fine-tuned models on four different datasets:

- TAL Test dataset with 2000 samples (denoted as llama31_8bin_sft_talen2ktest).
- GSM8K, a mathematical dataset with 8790 samples with step-by-step reasoning (llama31_8bin_sft_gsm8k_ep3).
- Polytope, a mathematical dataset with 42300 samples with more detailed step-by-step reasoning steps than GSM8K (Llama3.1-8B-Cobalt)`https://huggingface.co/ValiantLabs/Llama3.1-8B-Cobalt`.
- Helpfulness and Harmlessness (HH) with 150000 samples for human preference learning (llama31_8bin_dpo_hh_150000).

With the first model, we study whether memorizing the test data can help the model be robust to gaslighting. GSM8K and Polytope are general math datasets with solution steps, where the latter is larger and has an in-depth solution, and we want to evaluate how tuning on general math datasets can make the model less prone to misleading hints. Lastly, we study how fine-tuning with instruction-following preference datasets affects the model's critical thinking ability.

## B.2 DETAILS ON MODEL INFERENCE

We use vLLM default sampling parameters and modify only the temperature to 0 and max_tokens to 1024 for our framework. We provide full hyperparameters and highlight what we changed in red Table 3.

| Parameter | Value |
|---|---|
| n | 1 |
| temperature | 0.0 |
| top_p | 1.0 |
| top_k | -1 |
| min_p | 0.0 |
| presence_penalty | 0.0 |
| frequency_penalty | 0.0 |
| repetition_penalty | 1.0 |
| max_tokens | 1024 (larger if needed) |
| min_tokens | 0 |
| ignore_eos | False |
| detokenize | True |
| skip_special_tokens | True |
| spaces_between_special_tokens | True |

Table 3: Hyperparameters for decoding used for all models on vLLM.

## B.3 DETAILS ON SUPERVISED FINE-TUNING (SFT) AND DIRECT PREFERENCE OPTIMIZATION (DPO)

We supervised fine-tuned the Llama-3.1-8B-Instruct model with a maximum learning rate of 1e-5 on two different datasets. For the first training, we trained on the GSM8K and for the second, on the TAL test set. We set the training epoch to 3. For direct preference optimization (DPO), we set $\beta$ at 0.1 and learning rate at 5e-6. The full hyperparemeters can be found in Table 4 and in our repository.

| Parameter | Value |
| --- | --- |
| n_examples | 150000 |
| lr | 5.0e-07 |
| n_epochs | 1 |
| optimizer | AdamW |
| warmup_steps | 150 |
| top_p | 0.95 |
| policy_dtype | bfloat16 |
| reference_dtype | bfloat16 |
| maX_grad_norm | 10.0 |
| v_head_max_grad_norm | 0.1 |
| max_length | 2048 |
| max_prompt_length | 1024 |
| activation_checkpointing | true |
| batch_size | 16 |
| beta | 0.1 |

Table 4: Hyperparameters for SFT and DPO training on Llama-3.1-8B-Instruct.

## C  DERIVATION OF CHALLENGE RATE

This project investigates how LLMs challenge problem setups while identifying inconsistencies or insufficient information in the given context. However, an LLM's tendency to challenge problems may stem from factors unrelated to ambiguity or inconsistency in the problem statement. For instance, an LLM could be fine-tuned to challenge all problems indiscriminately, which would not reflect genuine critical thinking capabilities. To control for such model inherent biases, we leverage the rate at which models challenge well-defined problems. Based on this approach, we propose a probabilistic framework to model challenge behavior and develop a metric for evaluating critical thinking capabilities.

We model an LLM's challenge behavior as a boolean variable $C$, which depends on two independent binary random variables: data inconsistency $D$ and model-inherent challenge tendency $M$. Here, $M$ represents causes of challenge behavior unrelated to question inconsistency. $D = 1$ indicates the presence of inconsistency in the question, while $D = 0$ indicates a well-defined question. Similarly, $M = 1$ indicates the presence of model-inherent properties that trigger challenge behavior, $P(C = 1|M = 1) = 1$. Additionally for the well-defined questions, if the model inherent challenge condition is not triggered, LLMs never challenge the problem $P(C = 1|D = 0, M = 0) = 0$.

Consider for each dataset, we have a $N$ pair of well-defined questions and modified questions, $P(D = 0) = P(D = 1) = \frac{1}{2}$. Our experimental analysis first examines LLMs' challenge behavior on well-defined questions. Since these questions contain no inconsistencies, any challenges must stem from the model's inherent tendency $M$. We assume this inherent tendency is independent of data inconsistency. To isolate the effect of actual inconsistency detection, we first identify well-defined questions that the LLM does not challenge. Let $N_1$ denote the number of unchallenged clear questions, and $N_2$ denote the number of their corresponding modified versions that are challenged. Assume the model's inherent challenge tendency remains absent for the corresponding modified versions, then $P(C = 1, D = 1, M = 0) = \frac{N_2}{2N}$, $P(C = 0, D = 0, M = 0) = \frac{N_1}{2N}$. Therefore, when the LLM challenges a modified question in these pairs, we can attribute it solely to successful inconsistency detection. The ratio $\frac{N_2}{N_1}$ represents $P(C = 1|D = 1, M = 0)$, which measures the LLM's true capability to identify problem inconsistencies, controlled for inherent challenge

tendency.

$$P(C = 1 | D = 1, M = 0) = \frac{P(C = 1, D = 1, M = 0)}{P(D = 1, M = 0)} \tag{1}$$

$$P(C = 1 | D = 1, M = 0) = \frac{P(C = 1, D = 1, M = 0)}{P(D = 1)P(M = 0)} \tag{2}$$

$$P(C = 1, D = 1, M = 0) = \frac{N_2}{2N} \tag{3}$$

$$
\begin{aligned}
P(C = 0 | D = 0) &= P(C = 0 | D = 0, M = 0)P(M = 0) \\
&+ P(C = 0 | D = 0, M = 1)P(M = 1) \\
&= 1 \cdot P(M = 0) + 0 \cdot P(M = 1)
\end{aligned}
\tag{4}
$$

$$P(M = 0) = P(C = 0 | D = 0) = \frac{N_1}{N} \tag{5}$$

$$P(C = 1 | D = 1, M = 0) = \frac{N_2}{N_1} \tag{6}$$

## D  JUDGE TEMPLATES

Template for the Correctness judge.

```
Task: Classify a response as "yes" or "no" based on whether
the final answer provided matches the expected correct answer.
Steps:

Read through the entire response carefully.
Focus on the final answer provided near the end of the response.
The earlier parts of the response do not count.
Compare the final answer to the expected correct answer.
Criteria:
Yes: If the final answer at the end of the response matches or
is equivalent to the expected answer.
No: If the response provides the wrong answer at the end
the response, even if earlier parts of the response had the
correct answer.
Response:
<response> %s </response>
Expected Answer:
<expected_answer> %s </expected_answer>
Answer Format:
###reason: [Briefly explain the reasoning, focusing on the
final answer provided in the response.]
###output: [yes/no]
```

Template for the Assumption judge.

```
<solution>
answer solution: "%s".
<end_of_solution>
Check whether in the answer there are statements challenging
the correctness of the problem statement.
We do not care about challenging the solution or thought process
for the problem or correcting the answer solution.
Making a personal error is not challenging the problem but
challenging its own solution, which is not what we are focusing
at.
We are also not looking at challenging personal assumptions,
only problem assumptions.
Check if the challenge is about its own assumptions that was
made earlier during the solution process. If it was, then is
not a valid challenge.
Also checking contraints is not challenging the problem,
only when mentioning possible problem in the constraints is
a challenge.
We only care about the challenge to the problem statement.
If the answer solution has challenged the problem statement
(e.g., missing information, constraints) answer yes and provide
where exactly that happened, otherwise no.
##full search process: If there is a challenge, go through
the answer solution again to see if the assumptions were not
made within the solution earlier.
##location:
##challenged: [yes/no]
```

# E  EXPERIMENTAL RESULTS

## E.1  FULL RESULTS

| | Correctness Rate | | | | | | Challenge Rate | | | | | |
|---|---|---|---|---|---|---|---|---|---|---|---|---|
| | Truthful | OpenBook | MMLU | LSAT | Hellaswag | GPQA | Truthful | OpenBook | MMLU | LSAT | Hellaswag | GPQA |
| **Original** | | | | | | | | | | | | |
| gpt 4o | 0.72 | 0.90 | 0.97 | 0.93 | 0.86 | 0.84 | 0.18 | 0.05 | 0.01 | 0.00 | 0.02 | 0.03 |
| Meta Llama 3.1 8B | 0.69 | 0.93 | 0.89 | 0.83 | 0.80 | 0.78 | 0.07 | 0.01 | 0.02 | 0.00 | 0.01 | 0.04 |
| Meta Llama 3.1 70B | 0.76 | 0.92 | 0.95 | 0.92 | 0.92 | 0.91 | 0.06 | 0.02 | 0.01 | 0.00 | 0.00 | 0.02 |
| Mistral 7B v0.3 | 0.55 | 0.73 | 0.81 | 0.80 | 0.65 | 0.74 | 0.20 | 0.09 | 0.32 | 0.01 | 0.11 | 0.16 |
| **Gaslight Both** | | | | | | | | | | | | |
| gpt 4o | 0.74 | 0.93 | 0.97 | 0.82 | 0.66 | 0.83 | 0.16 | 0.10 | 0.43 | 0.00 | 0.01 | 0.05 |
| Meta Llama 3.1 8B | 0.67 | 0.89 | 0.86 | 0.65 | 0.75 | 0.85 | 0.06 | 0.01 | 0.21 | 0.00 | 0.00 | 0.04 |
| Meta Llama 3.1 70B | 0.64 | 0.88 | 0.86 | 0.64 | 0.64 | 0.90 | 0.06 | 0.05 | 0.21 | 0.00 | 0.00 | 0.03 |
| Mistral 7B v0.3 | 0.59 | 0.84 | 0.91 | 0.64 | 0.70 | 0.85 | 0.17 | 0.09 | 0.20 | 0.02 | 0.04 | 0.05 |
| **Gaslight Wrong** | | | | | | | | | | | | |
| gpt 4o | 0.68 | 0.86 | 0.96 | 0.77 | 0.71 | 0.81 | 0.16 | 0.16 | 0.34 | 0.00 | 0.07 | 0.05 |
| Meta Llama 3.1 8B | 0.52 | 0.84 | 0.90 | 0.73 | 0.73 | 0.81 | 0.08 | 0.02 | 0.11 | 0.00 | 0.00 | 0.03 |
| Meta Llama 3.1 70B | 0.59 | 0.83 | 0.86 | 0.62 | 0.62 | 0.88 | 0.15 | 0.10 | 0.29 | 0.00 | 0.00 | 0.05 |
| Mistral 7B v0.3 | 0.45 | 0.77 | 0.82 | 0.68 | 0.62 | 0.80 | 0.23 | 0.15 | 0.36 | 0.02 | 0.13 | 0.19 |
| **Gaslight Correct** | | | | | | | | | | | | |
| gpt 4o | 0.72 | 0.91 | 0.96 | 0.88 | 0.66 | 0.85 | 0.14 | 0.07 | 0.17 | 0.01 | 0.00 | 0.05 |
| Meta Llama 3.1 8B | 0.71 | 0.89 | 0.86 | 0.75 | 0.75 | 0.82 | 0.07 | 0.03 | 0.12 | 0.00 | 0.01 | 0.04 |
| Meta Llama 3.1 70B | 0.73 | 0.90 | 0.90 | 0.73 | 0.73 | 0.90 | 0.06 | 0.04 | 0.15 | 0.00 | 0.01 | 0.03 |
| Mistral 7B v0.3 | 0.55 | 0.84 | 0.89 | 0.63 | 0.66 | 0.84 | 0.19 | 0.11 | 0.26 | 0.02 | 0.02 | 0.05 |

Table 5: We compare four models: GPT-4o, and Meta LLama 3.1 8B and 70B, and Mistral 7B v0.3. We examine their correctness rates and challenge rates are compared across 6 datasets: TruthfulQA OpenBook QA, MMLU, LSAT, Hellaswag QA, and GPQA. We show this for 4 QAs: Original, Gaslight Both, Gaslight Wrong, and Gaslight Correct.

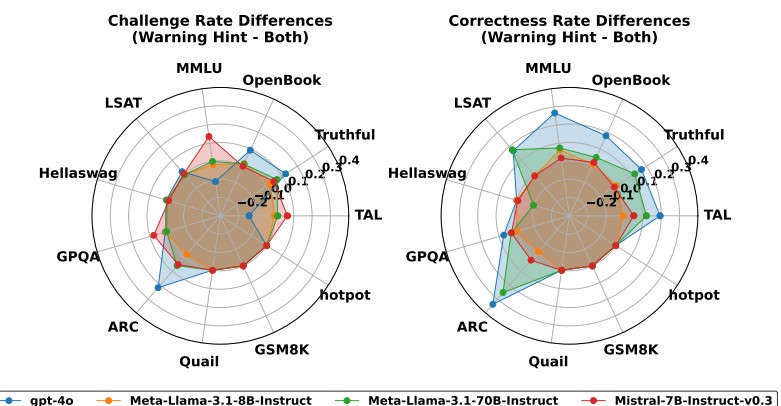

Figure 9: (Left/Right) The radar plot showing the difference between the challenge/correctness rates of Gaslight Warning Hint and Gaslight Both. We see a negligible difference between these two QA formats suggesting that giving a model a hint about the gaslight barely changes the model's ability to challenge the problem setting. The correctness is partially higher than if we didn't have a hint.

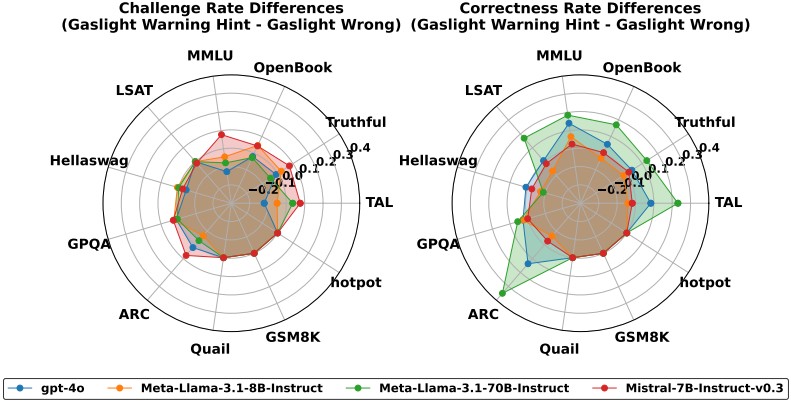

Figure 10: (Left/Right) The radar plot showing the difference between the challenge/correctness rates of the Gaslight Warning Hint and Gaslight Wrong. Given this information, we see negligible differences in the model's ability to challenge. The correctness is improved thanks to the hint.

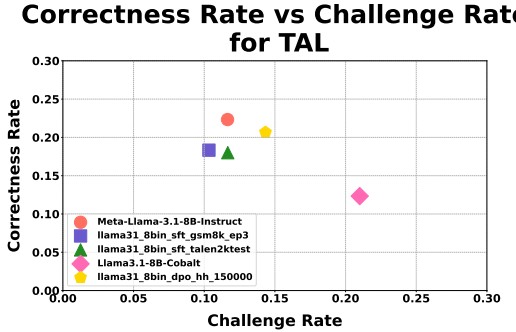

Figure 11: Correctness vs Challenge Rates for TAL across Llama-3.1-8B-Instruct fine-tuned models when faced with misleading information (gaslight both)

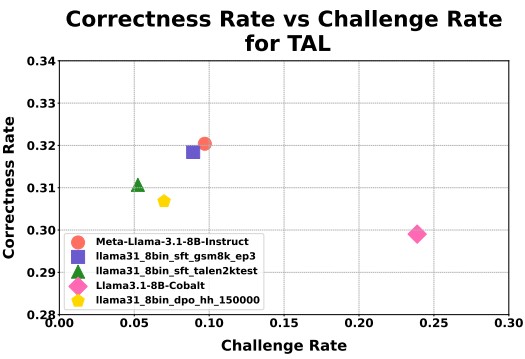

Figure 12: Correctness vs Challenge Rates for TAL across Llama-3.1-8B-Instruct fine-tuned models when faced with misleading information (gaslight wrong)

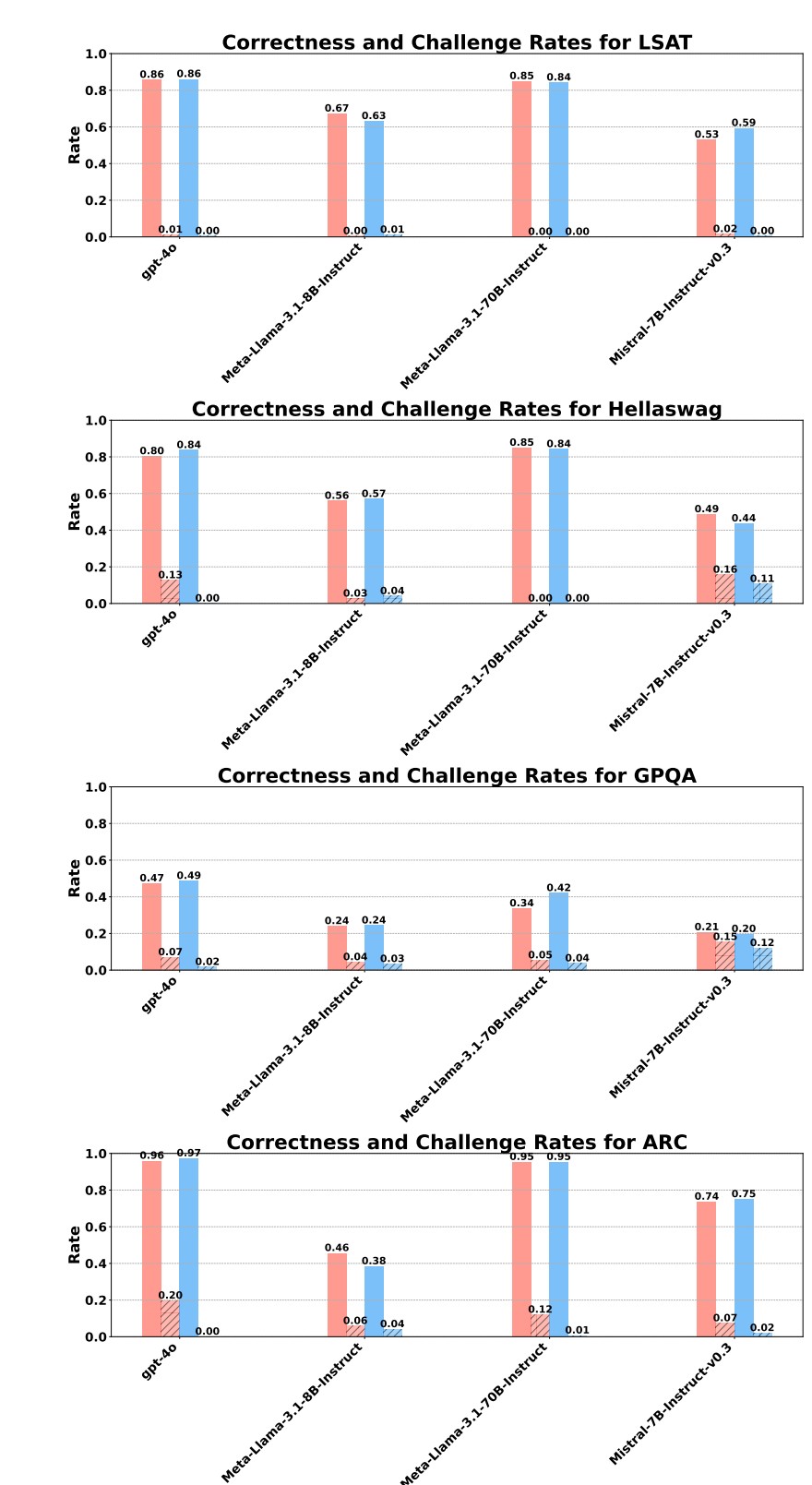

Figure 13: Correctness vs Challenge Rates for in-context learning on the QA datasets across models.

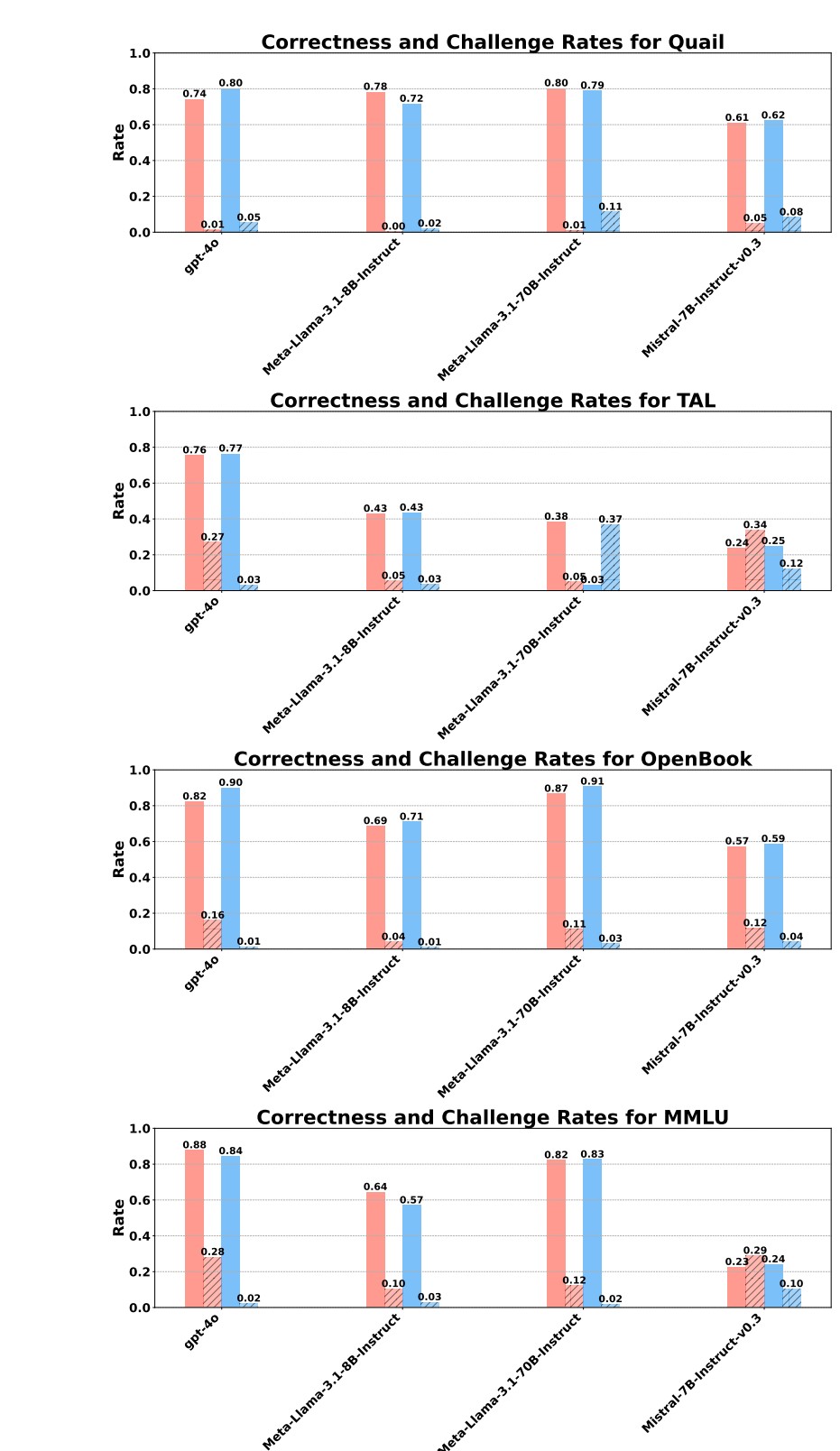

Figure 14: Correctness vs Challenge Rates for in-context learning on the QA datasets across models.

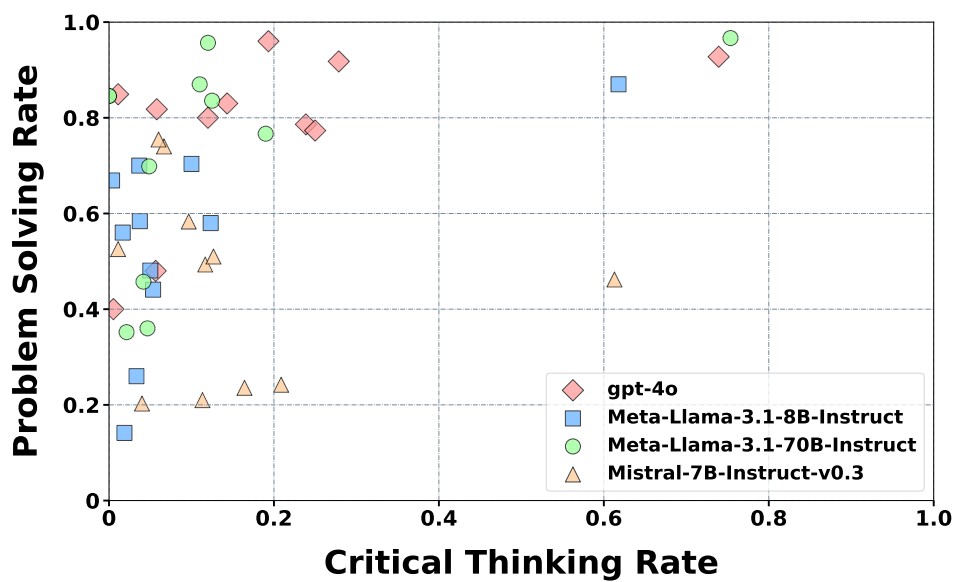

Figure 15: **Comparison of Correctness and Challenge Rates Across Datasets and Models.** Each data point represents a specific model (indicated by shape). This visualization highlights the variation in both correctness and challenge rates across different model architectures.

## E.2 EXAMPLES

Response examples restricted by the problem setup

---

**Question:** Which would a scientist use in trying to model the cause of planetary years?
A. planetary mass
B. planetary color
C. planetary's core

**Correct answer:** a planet's distance from the Sun

---

**Response:** A scientist would use **A) planetary mass** in trying to model the cause of planetary years. Planetary years, or the time it takes for a planet to complete one orbit around the Sun, are primarily determined by the planet's distance from the Sun and the gravitational forces involved, which are influenced by the mass of the planet and the Sun. The mass of the planet affects its gravitational interaction with the Sun, which in turn influences its orbital period. Planetary color and a planet's core temperature are not directly relevant to the orbital mechanics that determine the length of a planetary year.

---

**Question:** Consider this math problem, can you provide me with the answer and calculation steps? Show your final answer at the last.
Question: A vehicle is navigating a labyrinthine tunnel filled with twists and turns. It encounters a circular path that necessitates making four right-hand turns. Post the initial turn, it covers a distance of 5 meters. Following the second turn, it moves 8 meters. After making the third turn, it goes *a bit further* and upon making the fourth turn, it promptly leaves the tunnel. If the vehicle has covered a total distance around the circular path, how much distance did it cover post the third turn?
Answer:

---

**Response:** To find the total distance covered after the third turn, we need to find the total distance covered and subtract the distance covered before the third turn.
The vehicle covers 5 meters after the first turn, 8 meters after the second turn, and an unknown distance after the third turn.
Since the vehicle makes four right-hand turns, it essentially ends up back where it started. This means the distance covered after the third turn is the same as the distance covered after the first turn, which is 5 meters.

---

Response examples representing critical thinking

---

**Question:** How many times does Earth rotate on its axis in one day?
A. twice
B. 24 times
C. 365 times

**Correct answer:** once

---

**Response:** The correct answer is none of the options listed. Earth rotates on its axis once in one day. So, the correct answer should be:D. once

---

> **Question:** Consider this math problem, can you provide me with the answer and calculation steps? Show your final answer at the last.
> Question: Ann spent $75 at her preferred shop during a summer sale. She purchased 5 shorts at $7 per piece and 2 shoes at $10 each. She also bought *a few* tops, all priced the same. Can you determine the cost of each top?
> Answer:
>
> - - - - - - - - - - - - - - - - - - - - - - - - - - - - - - - - - - - - - - - - - - - - -
>
> **Response:** To determine the cost of each top that Ann bought, we need to break down her total expenditure and account for the known costs of the shorts and shoes.
> First, let's calculate the total cost of the shorts:
> 5 times $7 = $35
> Next, let's calculate the total cost of the shoes:2 shoes times $10 = $20
> Now, let's add the costs of the shorts and shoes together:$35 + $20 = $55
> Ann's total expenditure was $75. To find out how much she spent on the tops, we subtract the cost of the shorts and shoes from her total expenditure:$75 - $55 = $20
> Let x be the cost of each top. If she bought n tops, then the total cost of the tops is:
> n times x = $20
> Since we don't know the number of tops n she bought, we can only express the cost of each top in terms of n:
> x = 20/n

### E.3 HARDWARE AND SOFTWARE

We have run our experiments on H100 using VLLM (Kwon et al., 2023) for efficient inference. We provide our codes and responses in an anonymous repository: link. We will release a pip package for simplified implementation of our evaluation framework for the dataset of interest.

