# OpenReview forum: "LLM Spark: Critical Thinking Evaluation of Large Language Models"
_ICLR.cc/2025/Conference — Submitted to ICLR 2025_

### Official Review · Reviewer_JpKC · 2024-10-29

**Soundness:** 2
**Presentation:** 3
**Contribution:** 2
**Rating:** 5
**Confidence:** 3

**Summary:**

This paper introduces SPARK, a framework for evaluating Large Language Models' (LLMs) critical thinking abilities, specifically their capacity to identify inconsistencies in problem framing. The framework is grounded in the Hierarchical Three-Space Theory and evaluates LLMs across multiple dimensions (problem framing space, strategy space and implementation space) through five key hypotheses proposed by the authors. The authors create benchmarks by modifying existing datasets like commonsense QA, math and science datasets to introduce inconsistencies (e.g. missing options or missing conditions in the questions). Multiple LLMs are tested, and the experimental results show their limitations in critical thinking abilities.

**Strengths:**

- The framework is grounded in a cognitive theory (the Hierarchical Three-Space Theory)
- Extensive experimental results covering multiple domains and tasks, demonstrate the limitations of current LLMs in identifying inherent inconsistencies in provided problems.

**Weaknesses:**

- The findings that LLMs lack of ability to identify flaws and often agree with the hallucinations in the given queries are not surprising.
- It is unclear whether the modified question can truly capture the problem inconsistencies in the real world. It would be helpful to add a human baseline to see if this task is solvable and aligned.

**Questions:**

- How are the correctness rate and challenge rate calculated? Can a model that always rejects to answer questions obtain the highest challenge rate?

---

> ### Author Response · Authors · 2024-11-25
> **Thank you for your review**
>
> > **Findings that LLMs lack the ability to identify flaws is not surprising-agree with hallucination results…**
>
> We agree that the performance of critical thinking and hallucination have similarities. Here we want to clarify the distinction. Hallucination in LLMs usually refers to the cases when LLMs generate a response that is false, fabricated, or unsupported by the context. Within the 3-space solving theory framework, hallucination presents in the implementation space. Additionally, one of the causes of hallucination is the lack of knowledge or problem-solving capability. Our work evaluates model correctness to assess problem-solving capability and compare the performance of the modified question. In this way, we have a clear observation about how problem setup influences LLM behavior when relevant knowledge is available. According to our definition of critical thinking in the introduction section, we consider undesired behaviors, such as  selecting the wrong option or hallucinating the final result, indicate a lack of critical thinking. LLMs may produce inconsistent responses despite having relevant knowledge, as they rigidly adhere to question-answering structures in the model space. A critical agent should be able to recognize and declare when a problem is unsolvable through reasoning. In conclusion, our experimental design deliberately controls for knowledge insufficiency, demonstrating that critical thinking capability reflects a higher-order aspect of LLM behavior about how they approach problem-solving fundamentally. This distinguishes it from hallucination.
>
> > **Unclear whether the modified question can truly capture the problem inconsistency in the real world…**
>
> We agree that insufficient information or inconsistency from real-world users is usually more complex than our setting. However, our approach using established benchmarks provides a controlled environment to introduce ambiguity and develop reliable automatic evaluation templates for correctness and challenge rate measurement. These benchmarks, each targeting specific problem-solving skills, allow us to systematically investigate which types of tasks elicit critical thinking in LLMs and how task similarity influences performance (Section 4.6). The complexity varies across our datasets: modified GSM8k problems may appear obvious to humans, HotpotQA's implicit condition removal presents more subtle detection challenges, and multiple-choice questions become trivial when ground-truth answers are known. Despite some inconsistencies being relatively straightforward in our problem setup, LLM performance is not satisfying. This suggests that their performance would likely deteriorate further when faced with more complex or realistic queries where inconsistencies are more implicit. Importantly, our selected datasets span multiple domains and complexity levels, incorporating both common sense reasoning and domain-specific scientific knowledge.
>
> > **How are the correctness and challenge rates calculated?...**
>
> Correctness and challenge rate are calculated through the dataset. The correctness rate measures the proportion of responses demonstrating accurate knowledge. The challenge rate measures how often the model questions problem solvability. The automatic evaluation templates are displayed in Appendix D.
>
> For the second question, we have improved our critical thinking evaluation metric by incorporating challenge rates on well-defined questions. Consider for each dataset, we have a N pair of well-defined questions and modified questions. Our experimental analysis first examines LLMs' challenge behavior on well-defined questions. Since these questions contain no inconsistencies, any challenges must stem from the model's inherent tendency.. We assume this inherent tendency is independent of data inconsistency. To isolate the effect of actual inconsistency detection, we first identify well-defined questions that the LLM does not challenge. Let N1 denote the number of unchallenged clear questions, and N2 denote the number of their corresponding modified versions that are challenged. Assume the model's inherent challenge tendency remains absent for the corresponding modified versions. Therefore, when the LLM challenges a modified question in these pairs, we can attribute it solely to successful inconsistency detection. The ratio N2/N1 measures the LLM's true capability to identify problem inconsistencies, controlled for inherent challenge tendency. The detailed explanations are provide in the Appendix of the revised paper.

---

> ### Author Response · Authors · 2024-11-29
> **Seeking Your Additional Feedback on Paper12877 Revisions**
>
> Dear Reviewer JpKC,
> Following our responses to your thoughtful review of Paper12877, we would welcome your perspectives on whether our clarifications have sufficiently addressed your concerns, particularly regarding:
> - The distinction between hallucination and critical thinking in our framework
> - Our approach to evaluating real-world inconsistencies through controlled benchmarks
> - The enhanced evaluation methodology for challenge rates and correctness
>
> With the rebuttal period closing on December 2nd, any additional insights would be valuable in strengthening our manuscript.
>
> Best regards, Authors of Paper12877

---

> ### Author Response · Authors · 2024-12-02
> **Final Day - Paper12877 Feedback**
>
> Dear Reviewer JpKC,
>
> As the rebuttal deadline is today (Dec 2nd), we'd appreciate your feedback on our earlier responses to Paper12877.
>
> Best regards,
> Authors

---

### Official Review · Reviewer_pRhJ · 2024-11-02

**Soundness:** 3
**Presentation:** 3
**Contribution:** 3
**Rating:** 8
**Confidence:** 3

**Summary:**

Based on the three-space theory, this paper presents the SPARK hypothesis and assessment framework on LLM critical thinking. Through the benchmark constructed in the paper, the authors explored the current critical thinking ability of LLM, with the influence of various factors on it, through a large number of experiments, contributing to the assessment and enhancement of LLM's critical thinking.

**Strengths:**

Focusing on LLM's critical thinking skills, this paper frames the Benchmark by designing inconsistencies in the problem and designing a large number of experiments to explore it.

I personally like the idea of this work. In my opinion, the main strengths of this work include:
1) This paper uses the three-space theory to model LLM's critical thinking ability and explores the reverse proof of the framework, which provides a theoretical basis for critical thinking related research.
2) This paper conducts a large number of experiments to explore LLM's critical thinking and its influencing factors from multiple perspectives, which provides a feasible direction for the subsequent research.

**Weaknesses:**

I do not find significant shortcomings of this work, but only a few minor points to be clarified:
1. The critical thinking assessment was designed without taking into account the impact of the model's capabilities. For example, if the model itself cannot understand or does not have knowledge of the question, it is difficult to "criticize" it. This is especially true for multiple-choice questions and smaller models, as shown in Figure 2, where multiple-choice questions have a low percentage of correct answers and most models have a low change rate. This may result in an underestimation of their critical thinking, i.e., it is not that they do not have the ability to think this way, but that the questions are beyond their knowledge and ability.
2. In terms of assessment metrics, it is best to minimize the use of LLM assessments, which can be costly. For example, for multiple-choice questions, can there be a simpler way of assessing correctness rate, making the benchmark easier to use?
3. Correctness rate is sometimes used in complete problems [line 278] and sometimes in incomplete problems, and is also expressed as "none of the options" in its definition (incomplete problem), which can be confusing when reading the experiments and results.
4. Why does the gaslight increase its challenge rate while decreasing its correctness rate? If it affects the correctness rate, i.e., LLM is misled by gaslight, shouldn't the reasoning not be challenged but follow the misguidance?
5. Some of the dots and text in Figures 2, 11, and 12 overlap, which is hard to read.

**Questions:**

See weaknesses.

P.S., I'm really curious how o1 would respond to such problems.

---

> ### Author Response · Authors · 2024-11-25
> **Thank you for your review (1/2)**
>
> > **Critical thinking assessment without taking into account the model’s capabilities..**
>
> We have refined our correctness and evaluation metrics, with detailed explanations provided in our response to Reviewer Dx3U. Our updated methodology now incorporates LLM performance on clear questions as a measure of problem-solving capability.
>
> > **Minimizing the use of LLM as a judge to minimize costs…**
>
> We agree that there is potential for simpler evaluation metrics for response correctness. However, our experimental setup presents unique challenges that require a more sophisticated approach. Since we modify the original questions and require LLMs to explain their reasoning, simple linguistic matching metrics like ROUGE are insufficient. Our evaluation process involves two key challenges: first, extracting answer-relevant sentences from responses, and second, handling diverse response patterns across different benchmarks. While designing dataset-specific rule-based algorithms is possible, leveraging GPT-4's natural language processing capabilities offers a more efficient and flexible solution. Although this approach incurs higher computational costs, our evaluation templates demonstrate strong alignment with human judgment on our manually validated subset of data.
>
> > **Correctness rate is sometimes used in complete problems…**
>
> Thank you for highlighting this limitation. We use correctness to assess whether LLMs incorporate the required knowledge for specific tasks. We acknowledge that our initial correctness evaluation template had limitations in assessing problem-solving capabilities for generative tasks, particularly when dealing with incomplete questions that lack ground-truth comparisons. To address this, we've introduced a refined correctness evaluation metric that considers LLM responses to both clear and modified questions. We now consider an LLM capable of solving the problem if it demonstrates correct knowledge or provides correct answers in either the clear or modified question scenarios. We will display the new results in the revised paper.
>
> > **Why does gaslighting increase the challenge rate?...**
>
> In our gaslighting experiment, we demonstrate that models can be easily misled to answer questions incorrectly when presented with a misleading hint, even if they initially solved the problem correctly without gaslighting, thus resulting in a significant drop in the correctness rate, as illustrated in Figure 5. Simultaneously, we observe an increase in the challenge rates when gaslighting is introduced. Misleading hints can influence LLMs to select incorrect options. When generating inference steps to support their wrong choices, the LLMs produce reasoning paths that contain counterfactual or flawed statements. The increased challenge rate in these cases suggests that when reasoning paths contain obvious errors or contradict common sense, LLMs are more likely to identify inconsistencies and challenge the problem setup or the provided hints. This demonstrates that LLMs exhibit critical thinking capabilities when the implausibility of their inference steps is obvious.
>
> > **Text and dots in Figure 2, 11, 12 overlap, which is hard to read ..**
>
> For these figures, we reduced the messy annotations and decided to emphasize more on the information by simplifying them into tables and separating them into different graphs.

---

> ### Author Response · Authors · 2024-11-25
> **Thank you for your review (2/2)**
>
> > **Can we assess o1 performance?...**
>
> We appreciate the reviewer's interest in o1. Our evaluation revealed several practical limitations with this model:
>
> The table shows that o1-preview has poor performance and rarely challenges the prompt, less than 15 assumptions per QA format. We suspect this is due to the hidden reasoning tokens. Additionally the model has incredibly poor correctness rates against our QA_gaslight_both and QA_hidden correct, showing an ~85% drop in correctness.
>
>
> However, the model required a high max_completion_tokens setting (10K) to generate consistent responses otherwise it would return no response and the responses were fixed at temperature 1, limiting control over generation parameters. Additionally, the testing costs were substantial ($100 for 810 prompts).
>
>
> Even though we used the small scale of datapoints, we have already observed that gpt-o1 can be easily fooled by misleading hint by following it, as we observe below:
>
> |    | model          | QA_Format          |  #Errors |   #Challenges |   #No Challenges |   #Correct |   #Incorrect |
> |---:|:---------------|:------------------|---------------:|----------------:|--------------------:|----------------:|------------------:|
> |  0 | o1-preview.pkl | QA_gaslight_both  |                    0 |              13 |                 257 |              34 |               236 |
> |  1 | o1-preview.pkl | QA_hidden_correct |                         0 |              14 |                 256 |              43 |               227 |
> |  2 | o1-preview.pkl | QA_original       |                         0 |               1 |                 269 |             230 |                40 |

---

> > ### Comment · Reviewer_pRhJ · 2024-11-26
> >
> > I appreciate the authors' responses. I will keep my original scores as they are already good enough.

---

### Official Review · Reviewer_xYtG · 2024-11-02

**Soundness:** 2
**Presentation:** 2
**Contribution:** 2
**Rating:** 5
**Confidence:** 4

**Summary:**

This paper introduces a novel approach to evaluating LLMs' critical thinking in identifying flaws in problem formulation. Grounded in Three-Space Theory, the authors reformulate existing datasets as critical thinking evaluation ones by removing correct answer choices (for multiple-choice QA datasets) or removing necessary conditions (for free-form generation datasets). They assess the "challenge rate"—the frequency with which LLMs, prompted to detect flaws, correctly identify issues, using GPT-4 for automatic YES/NO judgments. To further evaluate the model's robustness to misleading information in the problem formulation, the authors also augment QA datasets with hints ("gaslighting") on correct/incorrect answers or both.

Experiment results demonstrate that while some larger LLMs can achieve non-trivial challenge rates ($>50\\%$) on free-form generation tasks only, there remains substantial room for improvement. Notably, the challenge rate does not correlate with model accuracy, and chain-of-thought prompting yields inconsistent effects on both metrics across models and datasets. Although gaslighting increases challenge rates across models, it also reduces accuracy, highlighting LLMs' susceptibility to manipulation.

**Strengths:**

1. This paper investigates an interesting evaluation dimension on whether LLM can critique flaws in the prompted problem formulation, complementary to widely-used instruction-following and LLM reasoning benchmarks.

2. The experiment results can support the major claims of the paper.

**Weaknesses:**

1. **Limited Insight into Findings.** As this is an evaluation-focused paper, deeper analysis and implications of the results are the most important contributions. Many findings are presented as direct observations, often summarized by broad statements like "experiment results are influenced by dataset properties, models, training ...", "[prompting methods]  achieves mixed results", or "[model names] are vulnerable to manipulation in prompts". While these findings may hold, they resemble insights from prior work (Section 2) and align with expectations under a well-constructed evaluation framework. Although the paper emphasizes a unique "critical thinking" evaluation, it’s unclear what additional insights this approach offers beyond previous evaluation works.

2. **Clarity and Rigor in Experiment Design.** This paper reformulates existing datasets to build a new benchmark that focuses on evaluating LLM critical thinking, but several important experiment details are missing, or not rigorous enough. For example, the implementation of "Missing Information" for non-Quail datasets is not clearly defined, and criteria for identifying and removing "necessary conditions" (to make questions unanswerable) are unspecified.
The validity of the "LLM-as-judge" approach in this new critical thinking evaluation benchmark is not clearly explained, nor are the "held-out datasets" used in evaluations.

    Additionally, the exclusive use of instruct-tuned models raises questions about claims regarding instruction training effects (e.g., Line 272), as non-instruct-tuned models are not assessed.  Including control prompts where no flaws should be detected is also important to investigate potential false positive problems and prevent simple flaw-reporting models from skewing the results. Also, it seems a random baseline (possibly achieving 50% challenge rates) can beat most models in identifying problem formulation flaws, but there is no related explanation and analysis. Further problems are listed in the "Questions" section.

3. **Readability and Conciseness of Main Text.** Introducing the new "SPARK" framework and articulating hypotheses is understandably challenging. However, the overall paper organization, especially in Sections 3 and 4, could be improved for readability. The flow from Section 3.1 to Section 3.2 feels disjointed, and hypotheses are discussed in fragments across sections, which complicates their verification for readers. While the reviewer appreciates the efforts of putting an experiment summary in Section 3.4, it lacks grounding in detailed results, making the introduction feel verbose. Tightening the organization and streamlining explanations would improve the paper's clarity and coherence.

**Questions:**

Major problems are written in Weakness #2, and here are some less severe questions that need clarification or presentation advice:

1.  Why choose this particular set of questions? Why there is a focus on reasoning-focused datasets?

2. What are the decoding parameters for most models used in the experiments? Some datasets and models are sensitive to these hyperparameter decisions, so it should be clarified in the paper. There is no need to seek for best hyperparameter combinations, but for reproduction purposes, it is needed to know these experiment details.

3. How are checkpoints compared in Section 4.7 different? More importantly, what are they, and how they are related to the analysis in the main text?

4. Presentation Advice: The fonts and colors in many figures are hard to read or interpret, and some figures contain confusing legends and annotations (e.g., Figure 10).

---

> ### Author Response · Authors · 2024-11-25
> **Thank you for your review (1/2)**
>
> > **Limited Insight into Findings…**
>
> While we acknowledge that some of our analyses and the hypotheses presented in Section 3.2 share similarities with existing work, we argue that our Knowledge and Behavior Conditioning hypothesis offers unique insights into LLM problem-solving processes within the 3-space framework. Our in-context learning experiments (Section 4.8) reveal a unique tension: while in-context examples improve model accuracy across various datasets, they simultaneously decrease the challenge rate. This finding is particularly noteworthy because in-context examples help LLMs better understand questions and more frequently reference correct knowledge—factors that should make inconsistencies more detectable. Instead, we observe that these examples appear to reinforce the model's belief in the Problem Framing Space, making LLMs more confident in applying familiar solution structures while becoming less likely to question the completeness of problem setups. This observation has significant implications for future prompt design, highlighting the need to balance accuracy with critical thinking capability.
>
> Furthermore, we also want to emphasize that our problem definition differs from prior work. By deliberately modifying well-defined problems to create ambiguous or inconsistent questions, we achieve two significant advantages. First, we maintain clear and controllable conditions for understanding the source of inconsistencies. Second, we can definitively attribute performance degradation to limitations in critical thinking capability rather than knowledge insufficiency. This is because we specifically assess problem-solving capability against questions where the model demonstrates knowledge competence but is constrained by problem formulation.
>
>
> > **Clarity and Rigor in Experiment Design…**
>
> We apologize for any confusion regarding the creation of incomplete generative tasks. Let us clarify our methodology for each dataset (this information will be added to Appendix A.2)
>
> + Quail is a reading comprehension dataset and includes questions whose correct answer is “not enough information”. We directly sample some questions and corresponding paragraphs as incomplete reading comprehension tasks
> + GSM8k contains arithmetic problems, where the final answer is calculated by all the numerical conditions provided in the context. We design a reliable template to leverage GPT-4o to rephrase the problem context and remove one provided numerical condition. The detailed template is provided in Appendix A.1
> + HotpotQA is a multi-hop reasoning task, requiring information extraction from multiple documents. The dataset provides the indices of related documents and sentences. We create incomplete tasks by removing one relevant document from the required set
>
>
> We have refined our critical thinking evaluation metric by incorporating challenge rates on well-defined questions. Consider for each dataset, we have a N pair of well-defined questions and modified questions. Our experimental analysis first examines LLMs' challenge behavior on well-defined questions. Since these questions contain no inconsistencies, any challenges must stem from the model's inherent tendency.. We assume this inherent tendency is independent of data inconsistency. To isolate the effect of actual inconsistency detection, we first identify well-defined questions that the LLM does not challenge. Let N1 denote the number of unchallenged clear questions, and N2 denote the number of their corresponding modified versions that are challenged. Assume the model's inherent challenge tendency remains absent for the corresponding modified versions. Therefore, when the LLM challenges a modified question in these pairs, we can attribute it solely to successful inconsistency detection. The ratio N2/N1 measures the LLM's true capability to identify problem inconsistencies, controlled for inherent challenge tendency. The detailed explanations are provide in the Appendix of the revised paper.
>
> > **Readability and Conciseness of Main Text…**
>
> We sincerely appreciate the reviewer’s suggestion to enhance the readability and conciseness of our main text. We have made significant efforts to improve the structure of our work, including removing the unnecessary transition between Sections 3.1 and 3.2 and integrating the experimental proposals with the hypothesis to create a more cohesive narrative. Additionally, we refined the experimental details to support reproducibility and streamlined explanations to enhance the clarity and coherence of our findings.

---

> ### Author Response · Authors · 2024-11-25
> **Thank you for your review (2/2)**
>
> > **Why choose these set of questions? Why focusing on reasoning-focused datasets?**
>
> These datasets span diverse problem types—including mathematics, reading comprehension, domain-specific science, and story completion—each designed to evaluate specific problem-solving skills (detailed in Appendix B.1). While these datasets assess different aspects of LLM problem-solving, they share some common elements, enabling us to evaluate our Across-Domain Abstraction Hypothesis about the transferability of critical thinking between similar tasks(Sec 4.6). Furthermore, these datasets provide unique ground truth answers, which is convenient for us to evaluate whether the LLMs incorporate the required knowledge.
>
> We specifically focus on reasoning tasks as they align with our definition of critical thinking: the ability to analyze inference steps and update assumptions about problem completeness. This focus is crucial because reasoning paths and intermediate steps provide necessary feedback points, allowing us to evaluate whether LLMs exhibit inconsistencies during their inference process.
>
>
> > **What are decoding parameters?**
>
> We provided decoding parameters through the table in Appendix B.2 in vLLM and a link to the documentation.
>
> > **Explain checkpoints in Section 4.7…**
>
> We apologize for the confusing write-up of a paragraph. In Section 4.7, we evaluate the performance of the Llama-3.1-8B-Instruct on the challenging mathematical dataset, TAL, under the gaslighting setting. Observing, low correctness rate of the original model on the test TAL dataset, we study how fine-tuning affects the ability of the model. We evaluate fine-tuned models on four different datasets:
>
> + TAL Test dataset with 2000 samples (denoted as llama31_8bin_sft_talen2ktest).
> + GSM8K, a mathematical dataset with 8790 samples with step-by-step reasoning (llama31_8bin_sft_gsm8k_ep3).
> + Polytope, a mathematical dataset with 42300 samples with more detailed step-by-step reasoning steps than GSM8K (Llama3.1-8B-Cobalt).
> + Helpfulness and Harmlessness (HH) with 150000 samples for human preference learning (llama31_8bin_dpo_hh_150000).
>
> With the first model, we study whether memorizing the test data can help the model be robust to gaslighting. GSM8K and Polytope are general math datasets with solution steps, where the latter is larger and has an in-depth solution, and we want to evaluate how tuning on general math datasets can make the model less prone to misleading hints. Lastly, we study how fine-tuning with instruction-following preference datasets affects the model’s critical thinking ability.
>
> > **Fonts and colors in figures are hard to see, some has confusing legends and annotation (Figure 10)**
>
> We are grateful for your review and have improved the figure to be clear, readable, and moved some information into tables for clarity.

---

> ### Comment · Reviewer_xYtG · 2024-11-27
> **Thanks for the detailed responses**
>
> Thanks to the authors for the detailed responses, which clarifies some of my confusion (terminology, paper organizations, and metrics). I also see the revision brings significant improvement to the paper. However, after reading the responses, I am concerned about the necessity of introducing SPARK as a necessary framework. The authors highlight two main contributions of the framework: (1) in-context learning experiments (Section 4.8) and (2) new experimental controls and analytical angles. While these contributions are valuable, they do not appear to justify the necessity of introducing SPARK. Even without SPARK framework, the paper’s experimental analyses could stand independently, raising questions about the central contribution and framing of the paper. I encourage the authors to reassess the positioning and main contribution of the work. This is not to suggest that SPARK lacks value, but its necessity within the context of this evaluation-focused paper remains unclear.
>
> Additionally, I recommend double-checking the newly introduced content for compatibility with existing sections, as some parts seem hastily added and there are some minor formatting issues. My concerns about the scattered verification of hypotheses are only partially addressed. Finally, the authors should avoid making definitive claims such as, "We can definitively attribute performance degradation to limitations in critical thinking capability rather than knowledge insufficiency." It is still an open question how models process and utilize injected knowledge, and these conclusions may only hold under the current experiment setup of fine-tuning and out-of-domain evaluation.
>
> Based on these concerns, I am inclined to slightly increase my score to acknowledge the authors’ detailed revisions and significant improvements. However, I maintain a negative assessment of the current version. I encourage the authors to revisit the framing and contributions to maximize the paper's impact in future submissions.

---

> > ### Author Response · Authors · 2024-11-29
> > **Thanks for your valuable feedback!**
> >
> > Thank you for your thorough feedback on our revised submission. Your insights on SPARK's positioning and suggestions about definitive claims will be valuable for improving our work. We appreciate the time you've taken to provide such detailed guidance.

---

### Official Review · Reviewer_Dx3U · 2024-11-03

**Soundness:** 2
**Presentation:** 1
**Contribution:** 2
**Rating:** 3
**Confidence:** 4

**Summary:**

The paper introduces SPARK, a framework intended to assess the capability of LLMs to identify inconsistencies in problem framings using modified existing datasets. The authors come up with two metrics "Correctness" and "Challenge Rate" for the evaluation. They use the idea of Three-space theory from the cognitive science to come up with this framework. The dataset consists of different domains  such as math, science, comprehension etc. They also introduce perturbation's to the data to see the changes in the output given by the LLM and evaluate those responses and try to analyze the behavior of LLMs. While the work is interesting but there are many issue with this, right from writing to selection of data and evaluation metrics.

**Strengths:**

The work uses inspiration from cognitive science to come up with framework

They address an important aspect of LLMs which is critical thinking

Multiple models are considered for the work and comparison

Multiple hypothesis are tested in this work.

**Weaknesses:**

-> First most of the figures are poorly inserted, there could have been other type of figures chosen as most of the figures have the data overlapping and it's hard to interpret them.

-> The writing is poor, there is too many things and little details

-> While the related work is good there are many more work that are missing one of them is "Tree of Thoughts"

-> The datasets chosen for this work are diverse and contains many existing datasets, there is no mention of testing of data contamination given the models that are considered for this work have this data in their training data, also the dataset could have been better, I feel there are better datasets like Game24, or the one's mentioned in the related work are more relevant to this work.

-> It is mentioned that you create benchmarks in the abstract, I didn't clearly understand exactly what that meant.

-> A framework paper should be more detailed such that others can reproduce and compare their work to this, need more quantitative results.

-> Also there could have been more evaluations metrics rather than having just two of them and using them to test variety of hypothesis, this decreases the robustness of the results.

-> Most of the figures in the appendix were hard to interpret, more details on them is appreciated.

**Questions:**

None

---

> ### Author Response · Authors · 2024-11-25
> **Thank you for your review (1/2)**
>
> > **Figures are poorly inserted…**
>
> Thank you for this feedback. We have revised the figure notations and clarified the data points in the captions
>
> >  **Writing is poor…**
>
> Thank you for the reviewer’s valuable feedback. In response, we have added more details to our findings to provide greater depth and clarity. We have also enhanced the presentation of our results and streamlined the manuscript to improve readability and overall comprehension. We have updated our manuscript to reflect the changes.
>
> > **Related work is good but missing…**
>
> Thank you for your suggestion. We employ Chain-of-Thought prompting to elicit the intermediate reasoning steps from the LLM, allowing us to assess both the model's knowledge accuracy and any inconsistencies between its understanding and the problem constraints. In our revised paper, we will expand the discussion to include other advanced prompting techniques, such as Tree-of-Thought and Graph-of-Thought approaches, as relevant methodological references.
>
> > **Data contamination**
>
> While detecting data contamination poses challenges, we have implemented specific measures to mitigate its effects. Our study primarily examines LLMs' critical thinking rather than problem-solving abilities. For modified multiple-choice questions, memorization of correct answers is less relevant, instead, we analyze how problem setup constraints influence LLM behavior. In modified GSM8k problems, we remove key numerical conditions, making it impossible to derive the exact answer. Thus, even if a model has encountered the original problem, providing the correct result would indicate flawed reasoning rather than desired behavior. Regarding HotpotQA, we specifically excluded questions that GPT-4o could answer correctly without all the necessary documents, ensuring our question pool requires genuine multi-hop reasoning.
>
> We have tested our evaluation framework on Game24. In this setup, LLMs must construct mathematical expressions using basic operations and each provided number exactly once to reach 24. We observed low problem-solving performance across all tested models, with a high frequency of responses indicating problems were unsolvable. Unlike our other datasets, Game24 does not allow for clear parallel versions of questions. It is straightforward to construct number sets that make the task unsolvable, however, replacing a single number creates an entirely new problem. This fundamental change means we cannot guarantee that the modified version maintains comparable complexity to the original. When LLMs respond that answers cannot be determined for modified questions, it becomes impossible to distinguish between two scenarios: (1) the model's inability to solve the problem, and (2) the model's successful recognition of task inconsistency based on its ability to solve similar problems. Given these inherent limitations in controlling for problem complexity and disambiguating the sources of model behavior, we conclude that Game24 is not suitable for evaluating critical thinking capabilities. We include the experimental results here.
>
> > **We create a benchmark…**
>
> We apologize for any confusion. Our work introduces a framework for creating benchmarks rather than offering a single, fixed benchmark. This framework enables users to assess the robustness of a trained model on their dataset of interest by evaluating its critical thinking abilities and examining how well these capabilities hold up for a given skill.
>
> > **Framework paper should be more detailed…**
>
> We appreciate the advice on reproducing our results. To support this, we have included a link to the vLLM sampling parameters documentation and a table that highlights the parameter changes used in our experiments. In addition, we provide the code and implementation used to generate our results. Furthermore, we have included the templates for automatic evaluation, covering challenge rates, correctness rates, and problem modifications.

---

> ### Author Response · Authors · 2024-11-25
> **Thank you for your review (2/2)**
>
> > **More evaluation metrics …**
>
> Our previous correctness evaluation focuses exclusively on modified questions, where LLM performance may be constrained by the incompleteness of generative tasks and incorrect options in multiple-choice problems. To enhance the robustness of our evaluation, we have introduced additional correctness criteria focused on assessing whether LLMs possess and utilize the necessary knowledge for clear problems. We implement this through clear, free-form tasks. Specifically, we remove predefined options for multiple-choice problems to convert them to free-form tasks and use clear problem descriptions for existing generative tasks. We evaluate the problem-solving capability using the correctness of both the clear problems and modified questions.  If an LLM demonstrates correct knowledge in either scenario, we consider this evidence of proper knowledge acquisition and understanding. The new metric for problem-solving capability offers a more reliable assessment of LLMs' knowledge acquisition.
>
> Additionally, we have improved our critical thinking evaluation metric by incorporating challenge rates on well-defined questions. Consider for each dataset, we have N pairs of well-defined questions and modified questions. Our experimental analysis first examines LLMs' challenge behavior on well-defined questions. Since these questions contain no inconsistencies, any challenges must stem from the model's inherent tendency. We assume this inherent tendency is independent of data inconsistency. To isolate the effect of actual inconsistency detection, we first identify well-defined questions that the LLM does not challenge. Let N1 denote the number of unchallenged clear questions, and N2 denote the number of their corresponding modified versions that are challenged. Assume the model's inherent challenge tendency remains absent for the corresponding modified versions. Therefore, when the LLM challenges a modified question in these pairs, we can attribute it solely to successful inconsistency detection. The ratio N2/N1 measures the LLM's true capability to identify problem inconsistencies, controlled for inherent challenge tendency. The detailed explanations are provided in the Appendix of the revised paper.
>
> > **More details on figures …**
>
> Thank you for your advice. We have made revisions to the visualization of our figures. We have enhanced the clarity of figure captions.

---

> ### Author Response · Authors · 2024-11-29
> **Looking Forward to Your Feedback on Paper12877 Revisions**
>
> Dear Reviewer Dx3U,
>
> We would appreciate your feedback on our thorough responses to your concerns about Paper12877. We have specifically addressed:
> - Enhanced figure clarity with improved notations and captions
> - Added substantial details to improve writing and readability
> - Expanded related work to include Tree-of-Thoughts
> - Detailed explanation of data contamination mitigation and Game24 testing
> - Added new evaluation metrics including challenge rates on well-defined questions
> - Clarified our framework's benchmarking capabilities
> - Improved reproducibility with detailed implementation documentation
>
> As the December 2nd deadline approaches, your input on whether these revisions adequately address your concerns would be valuable.
>
> Best regards, Authors of Paper12877

---

> ### Author Response · Authors · 2024-12-02
> **Final Day - Paper12877 Feedback**
>
> Dear Reviewer Dx3U,
>
> As the rebuttal deadline is today (Dec 2nd), we'd appreciate your feedback on our responses addressing figure clarity, writing improvements, expanded related work, and enhanced evaluation metrics.
>
> Best regards,
> Authors

---

### Author Response · Authors · 2024-11-25
**Dear Reviewers**

Dear Reviewers,


We would like to thank you for your valuable reviews, which have greatly contributed to improving our work. We are delighted that our research has been recognized as addressing a crucial aspect of large language models (Dx3U, JpKC)—critical thinking supported by the theoretical foundation of the Hierarchical Three-Space Theory (pRhJm, JpKC). Our claims are reinforced by experimental results (xYtG, pRhJ) and highlight promising directions for future research (pRhJ).

Based on your suggestions, we have made the following significant improvements to our work:

1. We **updated our figures** to **improve clarity and readability**. While our initial focus was on showcasing patterns, we overlooked their accessibility to readers. To address this, we refined plots and converted some figures into tables for better comprehension.
2. We **removed redundant information** and reorganized sections by combining related hypotheses. This restructuring eliminates disjointed parts and makes the narrative more cohesive and **easier to follow.**
3. We **expanded the reproducibility subsection**, ensuring that any reader can replicate our results. Additionally, we **released full code and datasets** to promote transparency and open research.
4. We **emphasized the key insights of our findings** and **refined our metrics** to highlight the primary message of our work. These changes also enhance the robustness of our results.
5. We **extended the related work section** to include advanced prompting techniques, providing a more comprehensive context for our contributions.

Finally, we want to underscore the broader significance of our work. Large language models are vulnerable to subtle changes, making it crucial to assess their critical thinking abilities to improve robustness. By introducing a straightforward framework for evaluating critical thinking metrics, we aim to provide model owners and users with actionable insights into model performance. We hope our work not only addresses this pressing issue but also inspires further research into enhancing the critical thinking and problem-solving capabilities of large language models.

We will be happy to address any further questions!

Kind Wishes, \
Authors of Paper12877

---

### Author Response · Authors · 2024-11-26
**We anticipate your feedback!**

Dear Reviewers,

With the rebuttal period coming to an end, we would greatly value your additional input. We want to express our sincere gratitude for the time and expertise you've invested in reviewing Paper12877 and helping us enhance its quality.

We would be particularly grateful if you could review our responses and indicate whether they adequately address your concerns, either fully or partially. We'd also appreciate knowing if our explanations are moving in a constructive direction.

Please don't hesitate to raise any additional questions or concerns about the paper. **As there is still time until our November 27th deadline, we welcome the opportunity to incorporate any changes that would strengthen the paper further.**

Best regards,
Authors of Paper12877

---

### Meta-Review · Program_Chairs · 2024-12-24

**Metareview:**

PC is entering meta-review on behalf of SAC and AC:

The reviewers felt that the paper's contribution was difficult to assess due partially to the writing/clarity of the work, and that there was limited insight that could be gleaned due to limited rigor in experiment design.

**Additional Comments On Reviewer Discussion:**

TBD

---

### Decision · Program_Chairs · 2025-01-22

Reject